**communications** engineering

# In vivo acoustoelectric neural recording in mice enabled by ultrasound-induced frequency mixing
Jean L. Rintoul ✉, Jonathan Howard, Patrycja Dzialecka, Xiaoqi Zhu & Nir Grossman ✉

There is a long-standing need in neuroscience for non-invasive methods that can record neural electrical activity with focal precision to diagnose brain disorders and interrogate circuit function. Here, we introduce acoustoelectric neural recording, which exploits ultrasound-induced frequency mixing to recover electrophysiological signals in vivo. Building on recent insights into the acoustoelectric interaction, we extend earlier work in cardiac tissue to demonstrate neural signal recovery in a living mouse brain. At the ultrasound focus, neural activity is shifted to frequencies near the acoustic carrier and can be retrieved by amplitude demodulation analogous to radio transmission. We further show that acoustoelectric neural recording is robust to artefacts and permits single-trial electrophysiological measurements. These results establish a pathway toward a real-time, portable, and non-invasive neural recording modality with the spatial precision of ultrasound.

There are currently no non-invasive techniques which can detect neural electrical activity in the brain with high spatial specificity and depth. Despite the importance of measuring electrical brain activity for understanding brain disorders and function current non-invasive neuronal recording techniques, such as electroencephalography[1] (EEG) are limited to diffuse cortical measurements of synchronised, large-scale events, where the electrical neural signals are strongly attenuated by the skull[2]. Magnetoencephalography[3] (MEG) can provide high spatial resolution measurements of electrical neural activity but only at the cortical layers. Blood-oxygenation-level-dependent (BOLD) functional magnetic resonance imaging[4] (fMRI) has three-dimensional spatial resolution but it is only an indirect and slow measure of changes in neural electrical activity.

There is a need for a technique capable of remotely isolating electrical neural activity with high spatial resolution and depth. The wavelengths of electrical fields are too long to achieve this via spatial interference[5]. Acoustic fields are capable of interferential focusing and if skull aberration[6] can be corrected for, focusing inside the human skull at depth is achievable[7]. Hence it would be advantageous to harness acoustic waves' focusing ability for the recording of neural signals.

The acoustoelectric interaction offers a viable means to detect electrical signals at the focal volume of the ultrasound. First discovered by Herzfeld and Rock in 1946[8] as a local conductivity change in the presence of an acoustic field, it has been utilised by Olafsson and Witte in ultrasound current source density imaging[9–11] (UCSDI) to detect ex vivo ECG signals ($\approx$1 mV in amplitude)[12]. However in vivo reports have been limited to the larger electrophysiological signals from the heart[13]. Recent investigations into the basic physics of the acoustoelectric effect have shown it is based on multiplication between the acoustic and electric field—aka heterodyning[14] such that a low frequency electric signal can shift up to the sum and difference frequencies around the acoustic carrier wave (Fig. 1), at only the spatial focus of the acoustic wave. Heterodyning is best known as the key technique behind radio communications[15] enabling low-frequency signals to be up-modulated by a carrier wave[16]. The original low-frequency signal can then be recovered by a demodulation technique such as Hilbert envelope analysis[17] or in phase and quadrature (IQ) demodulation[18]. Frequency shifting has numerous advantages—from minimising thermal noise which is largest at low frequencies[19,20], to being in a spectral band free of other neural signals as occurs with EEG. Thus acoustoelectric neural recording may address certain limitations of EEG, although it is likely to introduce challenges of its own.

In this work we harness the advantages of the acoustoelectric heterodyne interaction to provide narrowband demodulation of remote source signals eliminating other electrical artefacts from the signal recovery process. We then utilise the acoustoelectric heterodyne interaction to recover steady state visual evoked potentials (SSVEPs) in vivo the first time. Two artefact tests are used to evidence the mechanism behind the modulation a frequency specificity test to distinguish a heterodyne interaction and an acoustic isolation test to show dependence on the acoustic field. Lastly we show that single-shot acoustoelectric neural decoding is possible in spontaneous neural signals, providing the first in vivo evidence that a real-time acoustoelectric neural recording tool is possible.

Department of Brain Sciences, Imperial College London, London, UK. ✉e-mail: j.rintoul19@alumni.imperial.ac.uk; nirg@imperial.ac.uk

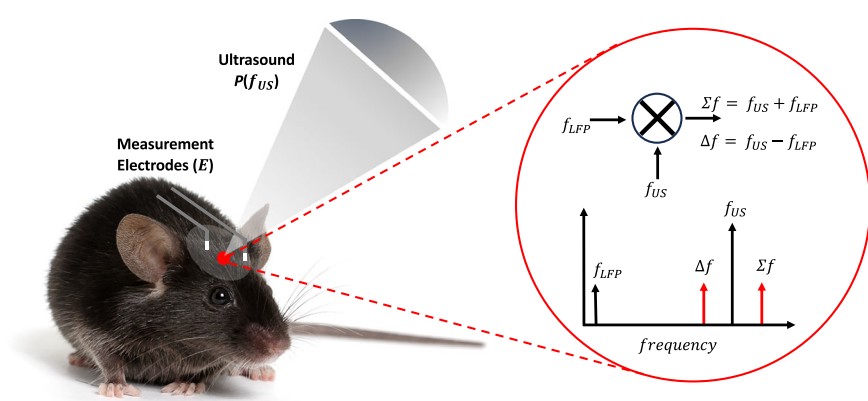

**Fig. 1 | Acoustoelectric neural recording concept.**
A low frequency endogenous electric field local field potential $E(f_{LFP})$ mixes with a highly focal acoustic signal $P(f_{US})$. The sum and difference frequencies emerge around the acoustic carrier signal through acoustoelectric heterodyning; $E(f_{LFP}) \times P(f_{US}) = E_{AE}(f_{US} \pm f_{LFP})$, where they can be recovered using amplitude demodulation. The depicted positions of the ultrasound focus and electrodes are illustrative and do not reflect the precise experimental configuration in the visual cortex. Mouse photo by Michiel de Wit (Standard License, Shutterstock).

## Results

### Acoustoelectric mixing enables minimisation of electric artefacts in signal recovery

To determine if the acoustoelectric heterodyne interaction could be used for in vivo neural recording we investigated the advantages of leveraging the heterodyne[14] components in remote signal decoding. Firstly the focality was verified within the electrophysiology arrangement using a 0.9% saline filled petri dish and a 1 V peak-to-peak 8 kHz sine wave voltage created from two platinum-iridium wires, with a 500 kHz continuous 1 MPa focused ultrasound transducer with water filled cone mounted above (Fig. 2a), within the intended electrophysiology instrumentation set up used for in vivo experiments (see 'Methods' and Supplementary Notes 1 and 2). By moving the stereotaxic instrument in 0.5 mm increments an XY map of the focal acoustoelectric field could be acquired at the sum (500 + 8 kHz) and difference (500–8 kHz) frequencies (Fig. 2b-i, ii) and a non-focal electric field was present at the acoustic frequency of 500 kHz (Fig. 2b-iii). The focal heterodyne products approximated the free-field acoustoelectric phantom studies performed with the ultrasound cone in place (Supplementary Note 3) and these acoustically focal fields cannot be explained by the mixing of electric fields alone. An artefact study investigating dependence on acoustic vibration at the electrodes was also performed, suggesting the medium is the source of the acoustoelectric signals and not vibration at the electrodes (Supplementary Note 4).

To assess the feasibility of demodulating in vivo neural signals in the low microvolt range[21] a 35 μV peak-to-peak 10 Hz sine wave voltage was applied to two platinum-iridium wires in 0.9% saline solution to simulate a neural signal and 500 kHz acoustic field with 1 MPa focal maxima positioned such that the focus targeted the electrodes (Fig. 2a). Here we expected to see the heterodyne products around the centre frequency however a large artefact at the centre frequency (Fig. 2c-i) dominated the spectral bandwidth obscuring any heterodyne products. To determine the source of the electric field at the acoustic frequency the electric field from powering the transducer was attenuated using an electrically insulating and acoustically transparent material, decreasing the electric field amplitude measured in the medium (see Supplementary Note 5a–c). By removing the ground plane of the transducer the electrical artefact was also increased (see Supplementary Note 5d–f), indicating the electrical artefact originated from powering the transducer which capacitively coupled to the medium[22]. To recover the small heterodyne products around this electrical artefact a Kaiser window (beta = 12) was applied to minimise spectral leakage[23,24]. Using this window sum and difference frequencies were revealed around the carrier i.e. 500,000 ($\Delta f = 500,000 - 10$, $\Sigma f = 500,000 + 10$) (Fig. 2c-ii). The Kaiser window enhanced the signal-to-noise ratio of the heterodyne products across 8, 12 s trials (Fig. 2d; $P = 2.99\text{e-}9$).

To determine if we could acoustoelectrically recover the time series neural amplitude electrical signal we compared a narrowband continuous wave to previously reported schemes which have included pulsing the ultrasound[13,25] to maximise the acoustic energy over a spectrally broadband.

In both cases the remote electric source signal was recovered through the same Hilbert envelope analysis which revealed small changes in the total energy of the measured waveform. The broadband approach reported in UCSDI[26] included the electrical artefacts from powering the transducer. We generated an 80 Hz PRF signal (50% duty cycle) at 500 kHz and compared the broadband (0.3–0.6 MHz) Hilbert envelope with the narrowband (500 kHz ± 50Hz) Hilbert envelope, computed over the same set of 80 Hz PRF trials. In the representative example (Fig. 2e), the recovered signals using the narrowband signal recovery had a higher mean correlation to the original 10 Hz source signal ($r = 0.51$; Pearson) compared to the broadband method ($r = 0.21$; Pearson). To enhance the amount of energy in the narrowband we moved to a continuous 500 kHz acoustic wave at 1 MPa, which resulted in a more accurate reconstruction of the original signal ($r = 0.68$; Pearson). Analysing over 8 trials the narrowband PRF signal recovery showed a higher correlation compared to broadband and the continuous wave narrowband signal recovery the highest correlation (Fig. 2f). Eliminating electrical artefacts from the signal recovery envelope and maximising the energy present in a narrowband enabled neural amplitude signals to be recovered in a saline phantom.

### Measuring neural electrical activity during concurrent continuous ultrasound

To determine if an acoustoelectric neural recording experimental paradigm was feasible we would need to apply continuous ultrasound and concurrently measure small microvolt range high frequency signals around the ultrasound transducer frequency as well as low frequency electrophysiological signals[27] (0–100 Hz). This approach differs from other experiments[28] in the field of ultrasound stimulation that use pulsed ultrasound stimulation[29–32] and comparatively low recording sample rates (5–100 kHz) with low-pass filters at the Nyquist frequency[33] to remove any higher frequency noise sources, such as the electrical artefact from the transducer.

Mice underwent a recovery surgery to enable the recording of visual evoked potential (VEPs) with platinum iridium electrodes placed in the visual and motor cortex, 7 mm apart from each other. Under light anaesthesia 0.5% (vol/vol) isoflurane in oxygen, VEPs were measured using a green LED stimulus positioned contralateral to the visual cortex electrode (see 'Methods' for surgery and experimental procedure).

Firstly an artefact free SSVEP response was established by shielding the LED from the mouse, to prove that the detected signal was not an electrical artefact from flashing the LED (see Supplementary Note 6). Then a green LED was flashed at 10 Hz with 50% duty cycle, while a 500 kHz continuous acoustic signal at 1 MPa was applied for 30 s inclusive of a 1 s onset and offset ramp and 1 s baseline and the in vivo electric signal recorded at a 2 MHz sampling rate. A front-end low-pass filter on the preamplifier (6 dB per octave roll off) removed all frequencies above the Nyquist frequency of the recording (1 MHz) to eliminate risk of aliasing and a front-end high-pass filter (0.1 Hz) added to reduce DC offsets so that the gain on the

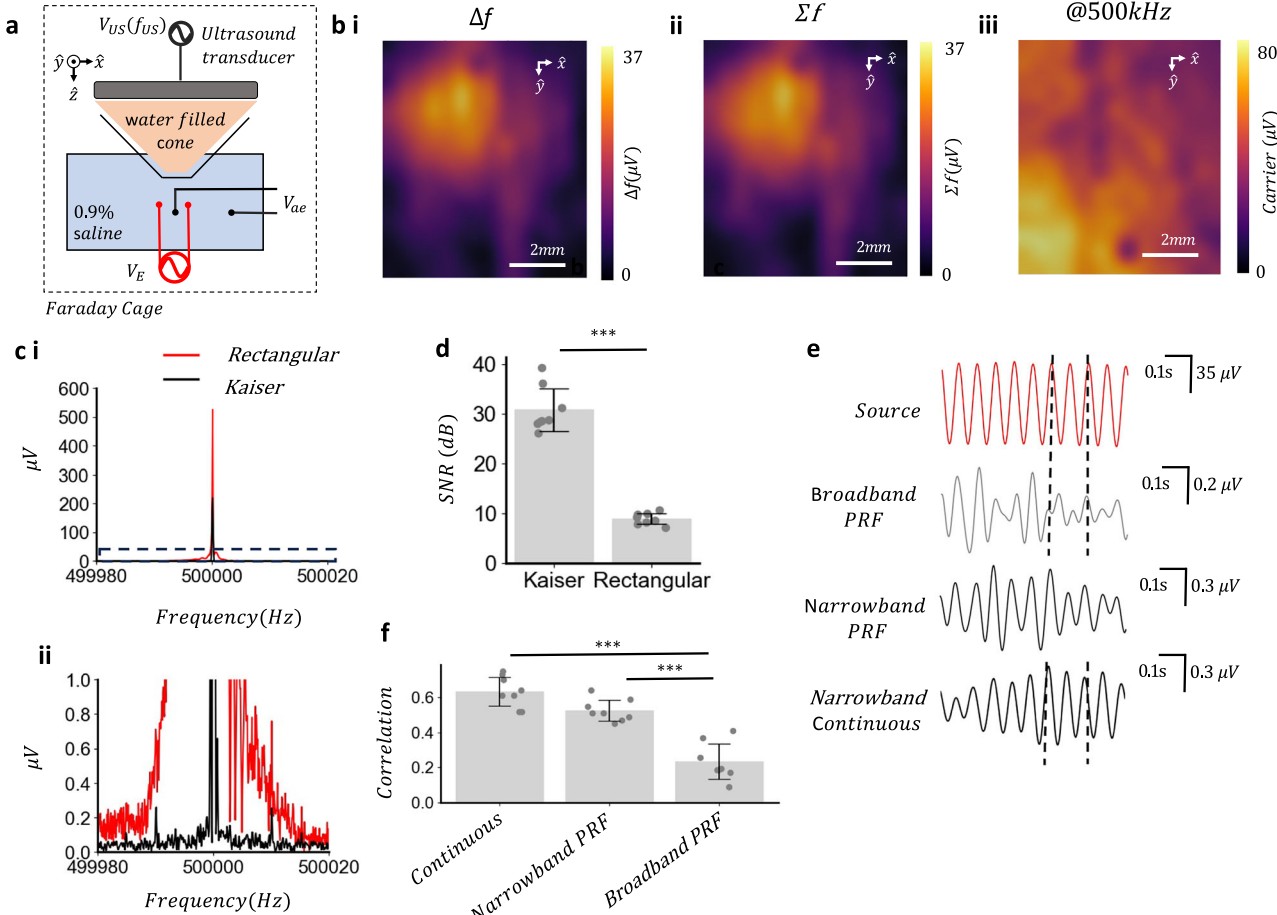

**Fig. 2 | Acoustoelectric heterodyning enables artefact free narrowband signal recovery of remote electric signals. a** 0.9% saline phantom with ultrasound cone positioned above with stimulation and platinum-iridium measurement electrodes spaced 7 mm apart used to create acoustoelectric field. **b** Acoustoelectric spatial map over 1 cm generated with 500 kHz acoustic wave at 1 MPa focal maxima 1 V electrical stimulation output at 8 kHz, moving in 0.5 mm increments. i $\Delta f$ amplitude map. ii $\Sigma f$ amplitude map. iii Carrier frequency (500 kHz) amplitude over same area. **c** Acoustoelectric amplitude comparison using a rectangular versus Kaiser window. Sampling rate 5 MHz and preamplifier gain at 5000 to measure $V_{ae}$, 12 s duration, 1 MPa focal pressure maxima@500 kHz and 10 Hz electric sinusoid ($V_E$) amplitude 35 μV measured at recording electrodes i Representative ASD comparison of signal measured at focus (Kaiser window beta = 12). ii Zoom view of (i) showing spectral spread of rectangular and the Kaiser window. **d** Signal-to-noise ratio (dB) comparison of $\Delta f$ and $\Sigma f$. Noise is defined as mean of ±2 Hz around $\Delta f$ and $\Sigma f$. t-test two sided $t_{(7)} = 13.12$, $P = 2.99e-9$; Kaiser group (mean ± s.d. = 30.81 ± 4.27 dB); rectangular group (mean ± s.d. = 4.27 ± 1.07 dB); $n = 8$ recordings. **e** Representative time-series of applied source signal (red) Hilbert envelope demodulation using broadband PRF signal recovery ($r = 0.21$; Pearson), narrowband PRF signal recovery ($r = 0.51$; Pearson) and continuous wave signal recovery ($r = 0.68$; Pearson), over a single trial. **f** Correlation between original source signal and recovered signals using continuous 500 kHz wave, narrow band 80 Hz PRF and broadband 80 Hz PRF envelope signal recovery approaches. ANOVA $F(2,8) = 44.40$, $P = 2.81e-8$, post-hoc comparisons using the Tukey HSD test, (continuous–narrow PRF, $P = 5.58e-2$; continuous-broadband PRF, $P = 2.80e-8$; narrow PRF-broadband PRF, $P = 4.05e-6$); over $n = 8$ trials.

preamplifier could be maximised without saturation. Each continuous ultrasound trial contained an electrical artefact at the acoustic frequency (500 kHz) with representative signal shown Fig. 3b and a DC offset starting at the onset of the continuous acoustic signal with representative signal shown in Fig. 3c.

To determine if continuous ultrasound could be applied while measuring an SSVEP continuous ultrasound trials were compared to baseline trials with no continuous ultrasound. Six mice, each with 5, 30 s trials with and without ultrasound were compared. The carrier signal was only present in the trials with ultrasound (Fig. 3d) ($t_{(29)} = 20.63$, $P = 1.89e-28$) as was the DC offset (Fig. 3e) ($t_{(29)} = 6.36$, $P = 3.38e-8$). The DC offset remained present in a similar set of continuous ultrasound saline phantom trials indicating the DC offset was not due to neural activity (See Supplementary Note 7). Then ultrasound was applied and an LED flashed at 10 Hz to evoke an SSVEP. The resulting neural signals were averaged and divided into 2 LED cycle sections with 95% confidence interval, revealing a repeating pattern (Fig. 3f). The amplitude spectral density (ASD) of a single 30 s trial (Fig. 3g) shows a clear 10 Hz fundamental with harmonics at 10 Hz intervals (Fig. S3d, f). The signal-to-noise ratio of the evoked SSVEP was then calculated for both the

ultrasound and no ultrasound groups (Fig. 3h) (US group mean ± s.d. = 20.91 ± 14.55 dB; no US group; mean ± s.d. = 19.05 ± 7.73 dB; $t_{(29)} = 0.60$, $P = 0.54$), indicating that SSVEPs can be measured during continuous ultrasound stimulation and are not significantly attenuated by the application of ultrasound, enabling the acoustoelectric neural recording paradigm to be tested in vivo. Front-end filtering (0.1–1 MHz) is used on the preamplifier to reduce the emergent DC offset at the onset of continuous ultrasound and eliminate the risk of aliasing from signals above the Nyquist frequency.

## In vivo acoustoelectric neural recording of visual evoked potentials

Now that we have developed a method to perform measurement of both VEPs in vivo and the electrical signals around the acoustic frequency with concurrent ultrasound stimulation, we attempted acoustoelectric neural recording using a 500 kHz focused transducer at 1 MPa. An 8 and 10 Hz LED SSVEP frequency was chosen to minimise the background thermally induced Johnson noise which follows a 1/f trend[20,27] and increase the spectral separation between the transducer electrical artefact at 500 kHz and expected heterodyned products compared to using a lower frequency

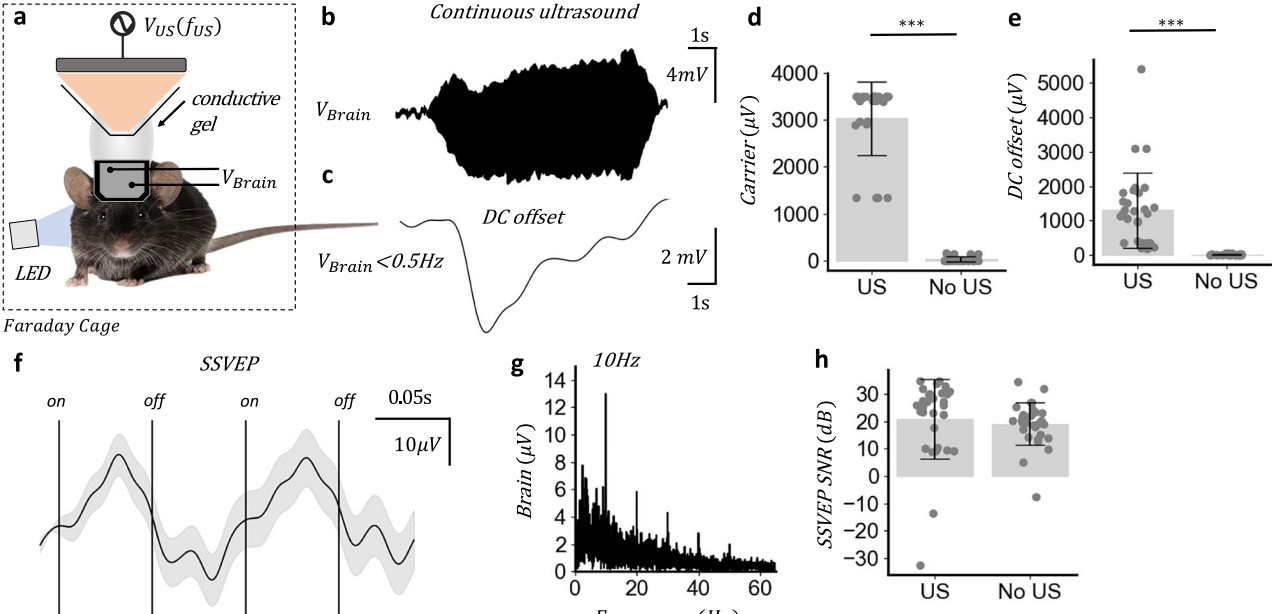

**Fig. 3 | Broadband electrical measurements during concurrent continuous ultrasound. a** In vivo acoustoelectric neural recording experiment with contralateral green LED stimulation and overhead ultrasound transducer with 500 kHz continuous sinusoid at 1 MPa, sampling rate = 2 MHz, preamplifier gain = 2000. Mouse photo by Michiel de Wit (Standard License, Shutterstock). **b** Representative raw recorded brain data without any filtering showing ultrasound transducer electrical artefact. **c** 0.5 Hz low pass filtered recorded signal. **d** Carrier amplitude comparison. $t_{(29)}$ = 20.63, $P$ = 1.89e-28; US group (mean ± s.d. = 3033.74 ± 778.92 µV); no US group (mean ± s.d. = 0.12 ± 0.09 µV); $n$ = 30 trials across 6 mice with 5 trials per mouse. **e** DC offset peak to peak height. $t_{(29)}$ = 6.36, $P$ = 3.38e-8; US group (mean ± s.d. = 1299.47 ± 1089.35 µV); no US group (mean ± s.d. = 11.82 ± 17.09 µV); $n$ = 30. **f** Ten hertz LED inducing a visual evoked potential averaged over 30 trials with 28 s of signal in each trial divided into non-overlapping 2 LED cycle segments, shown with 95% confidence interval error bar. All signals low pass filtered below 80 Hz. **g** Amplitude spectral density (ASD) over a single 10 Hz trial with continuous ultrasound shows 10 Hz VEP fundamental and harmonics. **h** SSVEP signal-to-noise ratio (dB) comparison of continuous and no ultrasound 10 Hz visual evoked potential paradigm. $t_{(29)}$ = 0.60, $P$ = 0.54; US group (mean ± s.d. = 20.91 ± 14.55 dB); No US group (mean ± s.d. = 19.05 ± 7.73 dB); $n$ = 30. SNR signal defined as ASD = 10 Hz compared to noise defined as mean of surrounding ±5 Hz spectral bins.

SSVEP. The ASD of the original visual evoked potential using an 8 Hz flashing green LED shows the fundamental 8 Hz and harmonics (Fig. 4a-i) and the Kaiser windowed[34] (beta = 12) spectral density around the carrier frequency at the sum and difference frequencies (Fig. 4a-ii). To differentiate the heterodyned signals from periodic broadband noise, we used an 10 Hz LED flashing frequency to evoke SSVEP's and performed a frequency specificity test (Fig. 4b) to differentiate the sum and difference frequencies from broadband noise near the acoustic carrier frequency averaging over 11 mice using 10, 30 s trials per mouse. The SSVEP amplitude varied between mice based on metabolic responses to Isoflurane anaesthesia[35]. By comparing the amplitudes around the acoustic carrier frequency (500 kHz) and a randomly selected and carrier (270 kHz) where no acoustic signal was applied, the SNR (dB) of the heterodyne products measured around the acoustic carrier frequency was significantly larger than the randomly selected carrier ($t_{(10)}$ = 4.37, $P$ = 0.0002). This frequency specificity test is necessary to determine that the mechanism behind the modulation is heterodyning with negative in vivo results evidenced when demodulation is performed at low frequencies which also contain periodic broadband noise caused by VEP harmonics. Artefactual demodulation can occur at low frequencies and the frequency specificity test should be used to exclude it (Supplementary Note 8). Furthermore demodulation under a pulsed ultrasound regime also makes neural responses susceptible to the auditory confound[36], suggesting that any acoustoelectric neural recording previously reported using pulsed ultrasound below 1 khz pulse repetition frequency is artefactually confounded unless appropriate confound tests are in place.

To recover the time series VEP the signal measured from the implanted brain electrodes was band filtered around the 500 kHz carrier frequency (±1 kHz) to remove all trace of the original low frequency VEP and its harmonics while leaving sufficient spectral space for filter ripple to be far from the tiny modulation products (Fig. 4c). In phase and quadrature (IQ) amplitude demodulation[18] was chosen over the net energy Hilbert envelope technique since IQ demodulation is more robust to carrier phase changes through heterogeneous media, reflections or non-coherent sampling. This gives IQ demodulation reconstruction advantages when there are complex multi-frequency signals[18] such as VEPs. IQ demodulation was then performed on the filtered signal to recover the original VEP from the heterodyned products. Since there was an electrical artefact transmitted from powering the transducer at 500 kHz this appears as noise below 5 Hz in the demodulated signal, thus the resulting signal was filtered between 5 and 40 Hz to remove the electrical carrier artefact. The demodulated signal was divided into segments based on LED markers in the recording and averaged to minimise the impact of spontaneous neural activity and repeated over 10, 30 s trials per mouse with a 95% confidence interval error bar, with the original VEP filtered similarly for direct comparison. The VEP and demodulated recovered signal averaged over a single mouse follow a similar trend shown in Fig. 4d with correlation ($r$ = 0.66 ± 0.14; Pearson). To generalise the correlation relationship, we repeated this test over 11 mice each with 10, 30 s trials per mouse, performing a frequency specificity comparison to the 270 kHz demodulated signal to prove the modulated signals were not broadband noise yielding a difference between the two groups ($t_{(10)}$ = 5.74, $P$ = 1.27e-5; Fig. 4e).

**Evidence frequency mixing is due to the acoustoelectric effect**
While we have shown that sum and difference frequencies are necessary to establish a heterodyne interaction has taken place mixing products could also occur due to electric only frequency mixing[5,37] with the electrical artefact from the transducer. To establish that frequency mixing was not due to electrical only mixing the acoustic signal was blocked by introducing an air gap at the end of the transducer cone, while conductive gel maintained electrical coupling to the side of the cone (Fig. 5a–i, ii). A green LED was

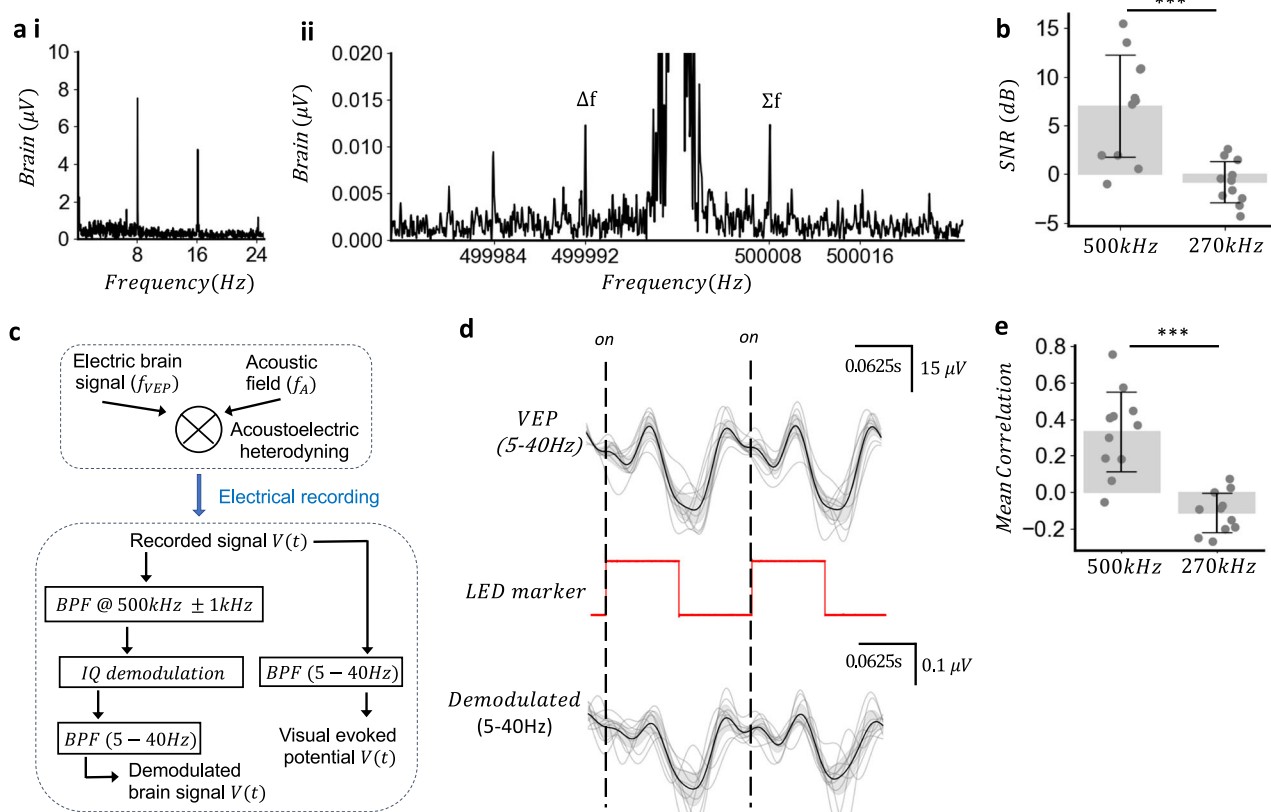

**Fig. 4 | In vivo acoustoelectric neural recording of visual evoked potentials.**
Acoustoelectric neural recording with an 8 Hz flashing LED and 30 s trials of
500 kHz continuous acoustic wave with 1 MPa focal maxima, brain electrodes
measurement preamplifier gain = 2000. **a-i** Representative ASD averaging 10, 30 s
trials from a single mouse, of the 8 Hz visual evoked potential (VEP). ii Kaiser
windowed spectrum (beta = 12) around the carrier frequency showing modulation
frequencies of VEP. **b** Frequency specificity test. Eleven mice with 10, 30 s trials
averaged for each mouse comparing frequency with (500 kHz) and without
(270 kHz) acoustic field. SNR (dB) calculated as signal = $(\triangle f + \Sigma f)/2$ and noise
calculated from the mean of the spectral bins ± 2Hz around the signal frequency. t-
test, two sided $t_{(9)} = 4.37$, $P = 0.0002$; 500 kHz group (mean ± s.d. = 6.99 ± 5.22 dB);

270 kHz group (mean ± s.d. = −0.78 ± 2.08 dB) across 11 mice. **c** Signal recovery
algorithm to recover the original signal from the heterodyned signal and separately
the VEP using in phase and quadrature (IQ) demodulation scheme. **d** VEP signal
averaged across 10, 30 s trials in a single mouse segmented into 2 LED cycles (average
in black, each trial shown in light grey, 95% confidence interval shown in shaded
grey); time correlated LED marker channel and IQ demodulated signal. Correlation
between VEP and demodulated signals r = 0.66 ± 0.14; Pearson. **e** Mean correlation
between the original VEP and demodulated signal. Demodulation performed at both
the acoustic 500 kHz frequency and 270 kHz. $t_{(10)} = 5.74$, $P = 1.27e-5$; 500 kHz group
(mean ± s.d. = 0.33 ± 0.22); 270 kHz group (mean ± s.d. = −0.11 ± 0.10);
$n = 11$ mice.

flashed at 10 Hz with the transducer acoustically connected, with repre-
sentative trial shown in Fig. 5b-i and the electrical carrier artefact measured
(Fig. 5b-ii) as well as the mixing products (Fig. 5b-iii). This was then
compared to the acoustically isolated case where the representative trial
(Fig. 5c-i) and equivalent electrical carrier measurement (Fig. 5c-ii) did not
induce equivalent modulation products (Fig. 5c-iii). This test was performed
over 7 mice with 5, 30 s trials for each mouse. The 500 kHz carrier frequency
amplitude varied minimally between the two groups (Fig. 5d, $t_{(30)} = -1.77$,
$P = 0.08$) and the VEP amplitude remained at similar amplitudes under
both conditions (Fig. 5e, $t_{(34)} = 0.34$, $P = 0.73$). There is a clear difference in
the signal-to-noise ratio of the heterodyne products (Fig. 5f, $t_{(34)} = 5.15$,
$P = 2.04e-6$), suggesting that the electrical artefact transmitted from the
transducer was not the dominant cause of the heterodyne products. The
sum and difference frequency products are dependent on their being a
transmitted acoustic field in accordance with acoustoelectric phantom
testing in Supplementary Note 3i. Since the carrier and VEP electrical
amplitudes between the two groups did not vary but the acoustoelectric
mixing products were different the source of mixing was not electric only.

To determine if electrical mixing could be occurring in the cabling
or preamplifier we see in the phantom spatial measurement (Fig. 2b)
that uses the same equipment electrodes and cabling as the mouse
experiment, the emergent field at the sum and difference frequency
varies with the spatial focality of the acoustic signal (Fig. 2b-i, ii),

indicating it is not due to mixing in the cabling or preamplifiers as this
would not induce an acoustically focal signal. Additionally0. we per-
formed a two-tone electrical mixing test at the same electrical ampli-
tudes and frequencies as measured in the in vivo experiments by
injecting two distinct electrical sinusoidal signals (tones) to assess the
linearity of the signal generation and recording system[37,38] (Supple-
mentary Note 9), providing further evidence that electrical mixing in the
recording system was not the dominant cause of the heterodyne pro-
ducts in in vivo experiments.

## Single-shot in vivo acoustoelectric neural recording of sponta-neous neural activity

Since the VEP amplitude is small averaging is usually deployed to reduce the
noise[39] (the signal but not the noise are time and phase locked to the VEP).
Single-trial acoustoelectric demodulation of neural activity would enable the
technique to be used as a real-time tool for recording spontaneous neural
activity. Using the same visual evoked potential experiment as described
previously we recorded spontaneous neural activity under low-level seda-
tion at 0.5% (vol/vol) isoflurane in oxygen such that spontaneous activity
could be measured (Fig. 6a) and recovered the neural signal through IQ
demodulation of a single trial.

To ensure the correlation in the demodulation was due to acousto-
electric heterodyning we employed both the frequency specificity and

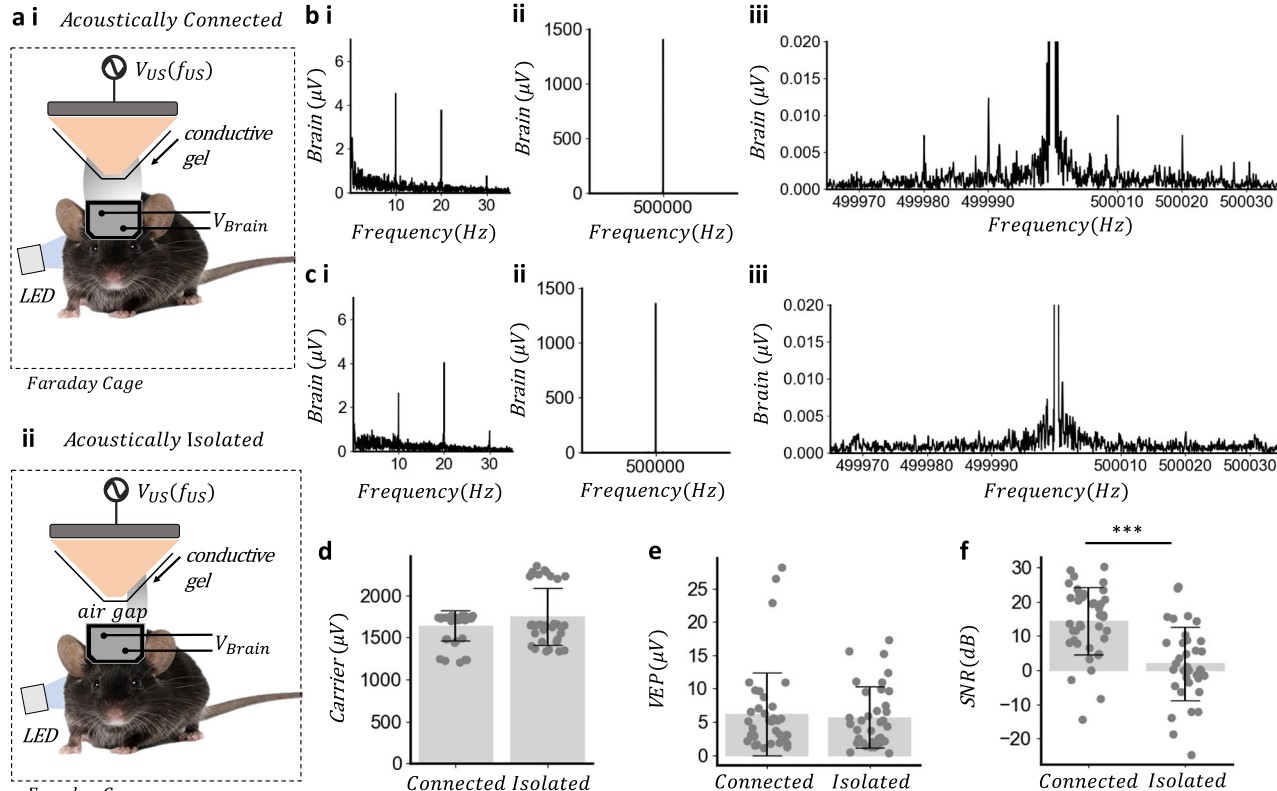

**Fig. 5 | Evidence frequency mixing is due to the acoustoelectric effect. a**-i Experimental arrangement for in vivo acoustoelectric neural recording preamplifier gain = 5000 with 500 kHz focused ultrasound transducer operating at 1 MPa, acoustically connected to the mouse via gel. ii Acoustically isolated in vivo experiment where the electrical connection is maintained by moving the ultrasound gel to the side of the cone, so that the mechanical wave is not transmitted through the air gap. Mouse photo by Michiel de Wit (Standard License, Shutterstock). **b**-i Acoustically connected representative visual evoked potential ASD acquired from 10 Hz flashing LED when ultrasound transducer provides 1 MPa continuously for single 30 s trial. ii Amplitude of the electrical carrier artefact from the ultrasound transducer. iii Spectrum showing the modulated 10 Hz VEP around 500 kHz carrier showing $\triangle f$ and $\Sigma f$, computed with Kaiser window (beta = 12). **c**-i Acoustically isolated representative visual evoked potential. ii Amplitude of the electrical carrier artefact is similar as in (**b**-ii). iii Spectrum where the modulated 10 Hz signal around

500 kHz carrier is expected computed with Kaiser window (beta = 12). **d** Carrier amplitude comparison between acoustically connected and isolated groups. t-test two sided $t_{(34)} = -1.77$, $P = 0.08$; Acoustically connected (mean ± s.d. = 1639.01 ± 178.71 μV); Acoustically isolated (mean ± s.d. = 1750.12 ± 341.12 μV); $n = 7$ mice, 5 trials in each mouse. **e** Visual evoked potential amplitude calculated at 10 Hz comparison between acoustically connected and disconnected groups. t-test, two sided $t_{(34)} = 0.34$, $P = 0.73$; Acoustically connected (mean ± s.d. = 6.14 ± 6.24 dB); Acoustically isolated (mean ± s.d. = 5.70 ± 4.54 dB); $n = 7$ mice, 5 trials each mouse. **f** Acoustic isolation test comparing signal-to-noise ratio (dB) between acoustically connected and isolated groups with SNR (dB) calculated as mean of $(\triangle f + \Sigma f)/2$, and noise as the mean of the spectral bins ±2 Hz around the $\triangle f$ frequency. t-test two sided $t_{(34)} = 5.15$, $P = 2.04e-6$; Acoustically connected (mean ± s.d. = 14.39 ± 9.89 dB); Acoustically isolated (mean ± s.d. = 1.92 ± 10.74 dB); $n = 7$ mice, 5 trials each mouse.

acoustic isolation test over 7 mice with 2 trials for each mouse, finding the mean correlation over the central 25 s of each 30 s trial between the onset and offset ramps. The acoustoelectric mean correlation across all trials was greater (mean ± s.d. = 0.43 ± 0.17) compared to the acoustic isolation test (mean ± s.d. = −0.03 ± 0.09); and frequency specificity test (mean ± s.d. = −0.00 ± 0.03); mean correlation (ANOVA $F(2,13) = 73.95$, $P = 1.73e-14$, Fig. 6b), revealing that acoustoelectric neural decoding can be performed without averaging. Focusing on a single acoustoelectric neural recording trial (Fig. 6c) the spontaneous neural activity (red) and demodulated signal (black) follow a similar path with some small variation, with mean correlation across the interval shown as $r = 0.72$, using a 0.5 s rolling window (Fig. 6d).

## Discussion
In this work we showed proof of concept that acoustoelectric neural recording can be achieved using acoustoelectric heterodyning in vivo, both using averaged SSVEPs and spontaneous neural signals. Two key artefact tests determine the mechanism behind the frequency modulation as acoustoelectric—a frequency specificity test to identify that a heterodyne process has occurred and an acoustic isolation test to show dependence on the acoustic field, as periodic broadband

electrical artefacts or electrical only mixing can both confound acoustoelectric signal recovery.

Because ultrasound can be focused through spatial interference at scales relevant to the human brain acoustoelectric neural recording has the potential to localise activity within small regions, both across cortical layers and at depth[40]. Unlike high-density ECoG[41] this approach may be implemented non-invasively, avoiding the surgical risks associated with electrode implantation[42] and unlike optogenetic methods—does not rely on transgenic modification of neural tissue[43]. By contrast fMRI relies on the BOLD signal, which reflects slow metabolic changes rather than direct electrical activity. With further development, acoustoelectric neural recording could provide access to previously unattainable information about brain function and may offer new avenues for diagnosing neurological and psychiatric disorders such as epilepsy[42], obsessive compulsive disorder[44] and depression[45].

Although the modulated acoustoelectric signal is smaller in amplitude than the EEG a key advantage of this approach is that it selectively shifts activity from neurons within the ultrasound focus to higher frequencies, isolating it from surrounding sources and thereby enhancing spatial specificity. Furthermore acoustoelectric neural recording has a different noise profile to EEG or MEG, with potential advantage that the thermal noise

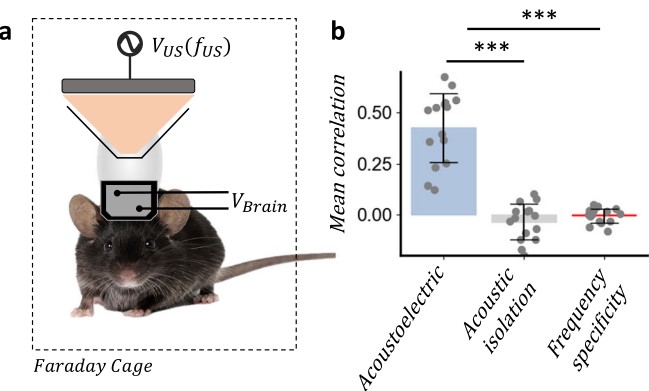

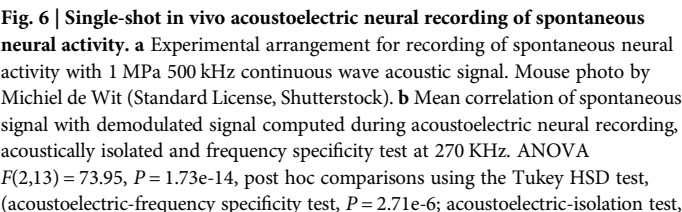

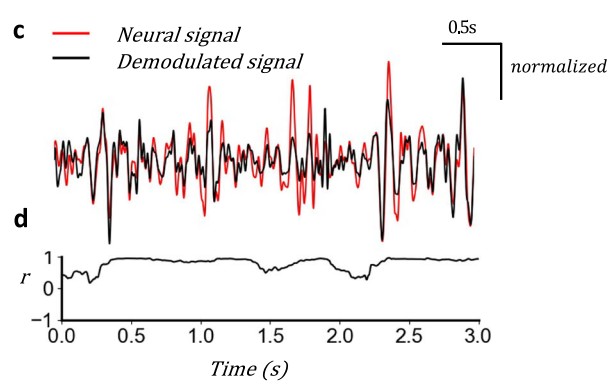

**Fig. 6 | Single-shot in vivo acoustoelectric neural recording of spontaneous neural activity. a** Experimental arrangement for recording of spontaneous neural activity with 1 MPa 500 kHz continuous wave acoustic signal. Mouse photo by Michiel de Wit (Standard License, Shutterstock). **b** Mean correlation of spontaneous signal with demodulated signal computed during acoustoelectric neural recording, acoustically isolated and frequency specificity test at 270 KHz. ANOVA $F(2,13) = 73.95$, $P = 1.73e-14$, post hoc comparisons using the Tukey HSD test, (acoustoelectric-frequency specificity test, $P = 2.71e-6$; acoustoelectric-isolation test, $P = 9.83e-7$; isolation test-frequency specificity test, $P = 0.76$), $n = 7$ mice with 2, 30 s trials for each mouse; Pearson correlation calculated over a 0.5 s rolling window. Acoustoelectric (mean ± s.d. = 0.43 ±0.17); acoustic Isolation (mean ± s.d. = −0.03 ± 0.01); frequency specificity (mean ± s.d. = −0.00 ± 0.03); **c** spontaneous neural activity (red) compared with demodulated signal (black), amplitude normalised for comparison, **d** rolling correlation (0.5 s window) over window shown in (**c**).

floor[20,46] which decreases with $\frac{1}{f}$, is much lower around the modulated frequency compared to the low frequency neural signals (see Supplementary Note 10). Compared to EEG the detection of the small acoustoelectric neural signals need not be limited to electrode position and size as the spatial locus is controlled by ultrasound, enabling larger electrodes to be used to average away any remaining noise[20,47].

Limitations in the experiments in this article are that we could not show focal specificity within the mouse model using a large wavelength (≈3 mm) 500 kHz ultrasound transducer due to emergent standing waves, which occur when the wavelength is a sub-multiple of the mouse head diameter (≈15 mm) with finite element simulations shown in Supplementary Note 11. Future work using a larger mammal or a smaller wavelength transducer may be able to overcome this limitation. While the recovery of spontaneous neural activity through single-shot acoustoelectric neural recording indicates the technique could be performed in real-time, the correlation between the original visual evoked potential and demodulated signal was limited, with varying amplitudes in the recovered demodulated waveform. These inaccuracies could be due to acoustic phase aberration due to the acoustic waves' propagation through heterogeneous media which could be overcome by the implementation of phase aberration techniques[48], standing waves induced by reflections creating periodic pressure changes, or non-coherent sampling caused by small timing differences between the signal generation and measurement hardware clocks[49].

A critical challenge for translating this technique to humans will be detecting the small acoustoelectric signal through the skull while operating within recommended ultrasound safety limits[50]. In this study we focused on establishing feasibility in a rodent model and assessing relevant artefacts. To simplify signal recovery we employed continuous-wave ultrasound, which facilitated frequency demodulation but also produced a DC-offset artefact that could be partially suppressed with high-pass filtering. Further systematic investigations in phantoms will be essential to determine the underlying cause of this artefact and develop methods for its mitigation. Although the mechanical index (MI) remained within diagnostic safety thresholds suggesting a low likelihood of cavitation, the associated temperature rise exceeded recommended limits (Supplementary Note 12). Therefore translation to human use will require optimisation of the acoustic waveform to maximise the signal-to-noise ratio of the modulated signal transmitted through the skull, while fully adhering to safety guidelines.

Acoustoelectric neural recording holds promise to become a spatially precise tool for non-invasive detection of neural signals, enabling improved insights into brain and nervous system function.

## Methods

### In vivo electrophysiology instrumentation
See Supplementary Note 1 for a diagram of the in vivo instrumentation arrangement and Supplementary Note 2 for electrode configuration comparisons between mouse and saline.

### Data acquisition
All applied signals were generated at 5 MHz and measured at 2 MHz using a 14-bit resolution analogue to digital converter, time synced using a CMI interface between the function generators and oscilloscopes with trigger inputs. The applied voltages monitor channels and electric potentials were logged using data acquisition hardware (WiFiScope WS5, WiFiScope WS6 DIFF and Handyscope HS5, TiePie engineering, Netherlands). The logged signals were streamed from the data acquisition hardware to a workstation PC using custom C code which utilised the Tiepie software development kit to control the triggers, channels and function generator output. The compiled C code which called the oscilloscope commands was in turn called within a Python wrapper.

### Electrophysiology brain recordings
The electrophysiological brain recordings were made by connecting a low-noise preamplifier with differential front end at 4 nV/√Hz input noise@1 kHz (SR560, Stanford Research Systems) to the implant in the mouse brain. All signals were measured using a 0.1–1 MHz band-pass filter on the SR560 to minimise any DC offsets induced by the DC bias at the beginning of recording and to remove noise above the Nyquist frequency.

### Pressure recordings
The acoustic field was measured using a 0.2 mm needle hydrophone (52 mV/kPa at 500 kHz calibration; model no: NH0200, Precision Acoustics Ltd.) and a DC-coupled preamplifier (Precision Acoustics Ltd).

### Acoustic fields application
The acoustic field was applied using a curved ceramic (PZT) 500 kHz ultrasound transducer (60 mm diameter, 63.5 mm acoustic path length; Precision Acoustics Ltd, UK). The ultrasound transducer was driven by an arbitrary function generator (Handyscope HS5, TiePie engineering, Netherlands) and a 40 W linear power amplifier (240L, Electronics and Innovation Ltd).

## Acoustic and acoustoelectric field cone characterisation

To understand the acoustic and acoustoelectric fields and their focality a full characterisation was undertaken in a more ideal 0.9% saline phantom environment which had fewer standing waves reflections induced by the either the mouse/air interface or the small petri dish. This system characterisation in a saline tank phantom with the cone attached over the ultrasound transducer is detailed in Supplementary Note 3, providing both acoustic and acoustoelectric measurements.

## In vivo electrophysiology

All mice were wild-type C57BL/6 male and female mice aged between 3 and 6 months. Mice were housed in standard cages in Imperial College London animal facility, with ad libitum food and water in a controlled light-dark cycle environment, with standard monitoring by veterinary staff. The Imperial College of London's Animal Welfare and Ethical Review Body approved all animal procedures and all experiments were performed in accordance with relevant regulations/according to the United Kingdom Animals (Scientific Procedures) Act 1986.

## Surgical preparation

Male(6) and female(5) ($n = 11$) C57BL/6 mice (3–6 months old) were anaesthetised with 2–3% (vol/vol) isoflurane in oxygen and received a dose of Carprofen (5 mg/kg), Buprenorphine (0.1 mg/kg) and saline for hydration (10 ml/kg/h) subcutaneously at the start of the surgery. The animal was head-fixed in a stereotaxic frame and the scalp removed. A custom-made head bar was attached to the skull with clear dental cement (PalaXpress Dental Cement, AgnTho's AB, Sweden) mixed with charcoal. Two holes (0.50 mm diameter) were drilled through the skull down to the dura with one hole in the V1 visual area (AP −3.50 mm, ML +2.25 mm) and the reference electrode in the motor cortex (AP 0.00 mm, ML −2.00 mm). Two custom-made electrodes (platinum-iridium 0.25 mm diameter, VWR, Lutterworth, UK) were placed in the holes and fixed to the head bar with cement. Low toxicity silicone sealant (Kwik-Cast, World Precision Instruments) was then poured around the electrodes to seal moisture around the craniotomy site and over the exposed skull, whilst also stabilising the electrode position. Nail polish was then applied to the exposed Pt-Ir wire to form a non-conductive exterior such that only the portion of wire in the brain was exposed (see Supplementary Note 2 for more detailed descriptions of the electrodes). This ensured measurements were from the brain and not above the brain in the ultrasound gel. Mice received Carprofen in water (5 mg/kg) for 2 days post-surgery with daily weight checks and allowed to recover for a week before experiments.

## In vivo electrophysiology experiment

Anaesthesia of mice was achieved in an airtight chamber at a constant 3% (vol/vol) Isoflurane in oxygen for 2 min after which the mouse was moved into the Neurotar head clamp ready for the experiment. Eye lubricant (Vaseline) was placed on the eyes to prevent them from drying out. The anesthetized mouse was placed onto a thermal mat (Thermostar, RWD, China) to maintain homoeostasis and head clamp (Neurotar standard clamp, Finland) to enable precise stereotaxic localisation of coordinates. The electrode implant was connected to the SR560 preamplifier (Stanford Research Systems, USA) via a small 1.27 mm header at the back of the head. The ultrasound transducer was mounted into a pre-characterised water-filled cone (see Supplementary Note 3). Parafilm was stretched over the end of the cone to seal in the water while the transducer was positioned via stereotaxic instrument above the mouse head covered in ultrasound gel (Anagel, UK) providing an acoustic connection between the cone and the mouse. Anaesthesia lowered to a constant 0.5% (vol/vol) Isoflurane in oxygen as visual evoked potential amplitudes were higher at lower anaesthesia levels. Ultrasound was applied in 30 s continuous trials at 1 MPa 500 kHz sinusoid, while a green LED was flashed in the eye contralateral to the visual cortex electrode implant. One megapascal was chosen to adhere to the established MI safety standards for ultrasound imaging[51,52] and to match

phantom feasibility experiments. The experiment was continued for an hour. After this time the mouse recovered so that it could undergo testing another day.

## Signal processing and data analysis

Custom Python scripts were created to using the numerical analysis coding libraries NumPy, Pandas and SciPy libraries.

Statistical tests. All data is shown as mean ± S.D. Statistical tests are specified in the respective figure legends. Statistical significance was tested using paired t-test and Tukey's honest significance test[53] corrected for multiple comparisons.

Frequency domain analysis. A 1-D discrete Fourier transform with flat top[54,55] window to optimise amplitude accuracy was computed, as we report ASD instead of power spectral density to easily read the average peak-to-peak amplitude of each measurement. After the Fourier transform is computed the two-sided amplitude spectrum was multiplied by 2 and half the array was taken—converting it into its single-sided form. The units of the single-sided amplitude spectrum then give the mean peak amplitude of each sinusoidal component making up the time-domain signal.

IQ demodulation algorithm for acoustoelectric neural recording signal recovery. First ultrasound was applied at 1 MPa 500 kHz for 30 s into the mouse head over the visual cortex whereby the acoustic field will mix with the endogenous electric signal evoked by the brain by the flashing green LED. The resulting electric signal is differentially recorded using two implanted platinum iridium electrodes at a 2 MHz sampling rate with a 14-bit ADC, with a hardware filter at the front end of the preamplifier from 0.1 to 1 MHz to avoid aliasing and high-frequency noise interfering with the recorded information.

To recover the modulated signal a 17th-order Chebyshev II infinite impulse response (IIR) analogue bandpass filter was applied from 499 to 501 kHz. This ensures no elements of the original low-frequency visual evoked potential were present in the band-passed resulting modulated signal. In IQ demodulation the received spectrum will be shifted downward and upward by the carrier frequency through multiplication with the in phase and 90° phase-shifted carrier and a low-pass filter is needed to suppress the high-frequency content at the sum frequency. The magnitude of the complex signal is computed where the square root of the sum of the $I^2$ and $Q^2$ components yield the magnitude of the resultant demodulated waveform.

Finally a 5–40 Hz bandpass filter was applied to the demodulated signal. The 5 Hz cut-off was implemented to remove the demodulated electrical artefact from the ultrasound which manifested around DC. The VEP frequency of 8–10 Hz was also chosen to avoid this low-frequency part of the spectrum where the electrical artefact from the ultrasound transducer dominates.

To directly compare the electrophysiological signal with the demodulated signal the same bandpass (5–40 Hz) was applied to the raw recorded signal to ensure symmetrical filtering on both the low-frequency electrophysiological signal and the modulated signal. After recovering both the resulting demodulated signal and VEP signal trial segmentation for averaging and correlation computations were undertaken and group statistics analyzed.

## Reporting summary

Further information on research design is available in the Nature Portfolio Reporting Summary linked to this article.

## Data availability

The data supporting the results in this study are available within the paper and its Supplementary Information. The raw and analysed datasets generated during the study are available for research purposes from the corresponding authors on reasonable request. A representative example of single-shot demodulation is available on Figshare at https://doi.org/10.6084/m9.figshare.c.7754537.

## Code availability

Example demodulation code on Figshare https://doi.org/10.6084/m9.figshare.c.7754537 and Github: https://github.com/Acoustoelectric/Acoustoelectric_Neural_Recording. Further information is available from the corresponding author upon request.

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

## Acknowledgements
N.G. was supported by the UK Dementia Research Institute (UK DRI)—an initiative funded by the Medical Research Council, Engineering and Physical Sciences Research Council (EPSRC) UK, Science and PINS Award for Neuromodulation, the NIHR IBRC Confident in Concept Award and the American Alzheimer's Association. We thank Prof. Robin Cleveland for signal processing advice, Dr. Nawal Zabouri for initial surgery training and Dr. Hazael Montanaro for advice on implementing acoustic finite element analysis in the software suite Sim4life.

## Author contributions
J.L.R. conceived and developed the hardware, experiments, analyses and wrote and revised the paper. J.H. developed the electrode implants and revised the paper. P.D. developed the original recovery surgery technique and revised the paper. X.Z. provided advice and training on electrophysiology and revised the paper. N.G. revised and edited the paper and provided supervision and project administration.

## Competing interests
The authors declare the following competing interests: a patent application WIPO(PCT): WO2024175933A1 was filed by Imperial Innovations.
