## [Transparent Peer Review File · Communications Engineering]

In vivo acoustoelectric neural recording in mice enabled by ultrasound-induced frequency mixing

Corresponding Author: Dr Jean Rintoul

Version 0:

Reviewer comments:

Reviewer #1

(Remarks to the Author)

Comments to the Authors

The authors present an interesting work that extends acoustoelectric measurements to in vivo neural recording. The topic of the paper is very relevant to the biomedical engineering community, and the use of ultrasound for enabling spatial selectivity and improved SNR in EEG measurements will be of interest and value to many scientists and engineers. In general, the paper is well-structured and well-written. Despite the acoustic parameter selection, which limits the capabilities of this work to demonstrate spatial specificity using ultrasound and the lack of non-invasive EEG measurements, it has sufficient merit to be published following the required corrections. However, the manuscript can significantly benefit from some clarifications regarding the methods used, as well as some editing (see comments and questions below).

General question: Throughout this work, you used particular ultrasound parameters of 0.5 MHz and 1 MPa. This parameter selection raises a couple of issues and, therefore, should be explained in the paper.

1. As mentioned in the discussion, this frequency limits your ability to demonstrate spatial selectivity in mice. Did you consider using a higher frequency? If so, why was this specific frequency selected?
2. Additionally, this relatively high pressure, combined with a low frequency, results in a relatively high MI (although still below imaging limits). Did you monitor for cavitation events or local temperature change?

A few comments and questions:

3. Please provide a more detailed explanation of the result in Fig. 3 h. This result was initially not intuitive to me, and readers could benefit from an explicit explanation of why we expect the SNR with and without US to be the same in this context.
4. How do you explain the temporal variability in the correlation presented in Figure 6C? It is not evident to me why the acoustic phase aberration due to the propagation of acoustic waves through heterogeneous media is time-dependent (line 287 in the discussion).
5. Figure 2a, please add the stereotaxic instrument to the illustration.
6. Figure 2c, please rephrase this sentence to make it more readable.
7. Line 432: Please add the model of the hydrophone.
8. Supplementary information: Figure S2 instead of Figure S8 in line 50.

Reviewer #2

(Remarks to the Author)

This study presents a novel method for recording neural activity in vivo by leveraging the acoustoelectric heterodyne interaction. The authors demonstrate, for the first time, the capability to detect both evoked and spontaneous brain signals at the focal point of a focused ultrasound field through frequency mixing with endogenous electric fields. Using a 500 kHz continuous wave ultrasound and in vivo rodent models, the team records steady-state visual evoked potentials (SSVEPs) and successfully demodulates neural signals via both Hilbert envelope and in-phase/quadrature (IQ) techniques. The authors validate the acoustoelectric mechanism using two key control paradigms: frequency specificity and acoustic isolation tests. Their findings indicate that acoustoelectric neural recording enables spatially resolved, artifact-resistant electrophysiological measurements even during continuous ultrasound exposure, offering a potential path toward non-

invasive, high-resolution neural recording techniques surpassing traditional EEG.

While the authors employ 1-second onset and offset ramping—a known technique to reduce auditory transients—the manuscript does not explicitly address the possibility of auditory confounding effects, particularly under the 80 Hz PRF condition used in some experiments. Rodents are known to exhibit neural responses to low-frequency ultrasound pulses via auditory pathways, raising the possibility that the recorded signals may partially reflect indirect auditory activation. Additionally, although the primary aim of the study is signal readout rather than neuromodulation, the use of 1 MPa continuous-wave ultrasound raises the possibility of direct neuromodulatory effects, especially in the spontaneous activity recordings. Without controlling for such effects, it remains unclear whether the recovered signals truly reflect pre-existing endogenous neural activity or ultrasound-induced changes. A more thorough discussion or additional controls (e.g., auditory masking, sham stimulation, or cortical monitoring) would help strengthen the attribution of the recorded signals specifically to acoustoelectric demodulation.

Although the text confirms the use of intracranially implanted electrodes for signal acquisition, Fig. 1 labels the measurement electrode as "external," which could be misleading. For clarity, it may be helpful to revise the label or caption to indicate that the electrode tip is intracortical while only the connector is external.

While the authors acknowledge the emergence of standing waves in the mouse model (Fig. S7), it remains unclear how these acoustic field distortions may have impacted the spatial specificity and fidelity of the recovered acoustoelectric signals. A more detailed discussion—or ideally, a sensitivity analysis—would help assess whether the measured heterodyne signals can be reliably attributed to the intended focal region under these acoustic conditions.

Although conceptually distinct, Yu et al. (2016, IEEE T-BME) reported noninvasive electrophysiological imaging of tFUS-induced brain activity using scalp EEG and source localization. Citing this work would help contextualize the current study within the broader spectrum of electrophysiological approaches to tFUS monitoring.

Reviewer #3

(Remarks to the Author)

This manuscript introduces and validates a method for recording in vivo neural signals through acoustoelectric heterodyning. The technique uses focused ultrasound to modulate endogenous electrical signals from the brain, allowing spatially specific electrical recording deep in the brain. The proposed approach could offer substantial advantages over EEG and MEG but some aspects require further clarification.

Major Points

- 1- The manuscript does not address a critical potential confound that the ultrasound itself, especially at 1 MPa continuous wave, may modulate neural activity. Both thermal and mechanical effects of ultrasound are known to influence excitability, synaptic transmission, and oscillatory dynamics. This raises
 - 1a. safety concerns and
 - 1b. the possibility that the recorded signals are not purely passive reflections of endogenous brain activity but may be altered by the recording method itself. I recommend the authors cite relevant LIFU neuromodulation studies, clarify whether any steps were taken to assess or limit neuromodulatory effects in the present setup (e.g., short durations, behavioral baselines, histology).
 - 1c. Pg. 4 line 147 - The manuscript states that the carrier-frequency and DC offset signals are only present during ultrasound trials. In order to distinguish whether signals are the result of passive detection or ultrasound-induced neuromodulation of neural activity, perhaps apply the technique to a non-evoked state (e.g., in anesthetized or postmortem tissue) to determine if the signal still appears, confirming it's not neuromodulatory in origin.
- 2- The ultrasound frequency used 500 kHz produces a focal region several millimeters wide, comparable to the entire size of a mouse brain. How can the method resolve or isolate neural activity from localized sources if the ultrasound interacts with multiple brain regions simultaneously? Clarify in the discussion how higher frequencies or larger animal models could solve the focality issue while overcoming attenuation of FUS and what steps are needed toward human translation.
- 3- The manuscript does not discuss how the implanted brain electrodes may interact with the ultrasound field. This is an important omission, as the presence of an electrode in the acoustic focus can introduce potential confounds, such as viscous heating artifact at the metal-tissue interface, acoustic scattering or reflection, micro-vibrations of the electrode, etc.
- 4- IQ demodulation assumes synchronization between the incoming signal and the reference oscillator (carrier). At 500 kHz, even a microsecond-scale delay may introduce significant phase error. This can potentially distort the I and Q components, especially when trying to track very slow (<20 Hz) modulations like neural signals. In addition, drifts in ultrasound amplitude or phase due to tissue inhomogeneity, motion, or coupling instability may occur. Please discuss whether these factors may limit the generalizability or robustness of the method in less controlled settings (e.g., freely moving animals or chronic implants).
- 5- Pg. 14 - The manuscript refers to the mechanical index (MI) to describe ultrasound safety and exposure, but MI alone is insufficient to characterize the acoustic field. Please include acoustic intensity (I_{spta} , I_{sppa}), thermal index and other relevant parameters.
- 6- Emphasize more clearly in the abstract and introduction how this work advances beyond prior UCSDI methods and recent acoustoelectric studies. Quantitative results in the abstract should also strengthen the work importance.
- 7- Discuss potential sources of residual error in the demodulated signals, e.g., phase aberration, as mentioned briefly in the discussion, and whether phase correction techniques might mitigate this.
- 8- Not all frequency choices are well justified, for example, in pg. 3 line 107, 80 Hz PRF, Hilbert enveloped +/-50Hz. A brief explanation should be added for why each frequency was chosen, what prior literature (if any) supports this, and whether

different frequencies were tested or ruled out.

Minor comments

1. Add brief comparative discussion of how this method fits among existing high-resolution neural recording techniques (e.g., high-density ECoG, optogenetic reporters, MEG).
2. Provide a conceptual roadmap toward non-invasive application in humans. This would highlight key technical gaps (e.g., deeper penetration, skull aberration correction, safe intensity limits).
3. Supplementary Figure S2 wrongly numbered.
4. Pg. 2 line 93 – Reference 23 seems misplaced. As written, the placement of the citation suggests that this result was reported by Reference 23, which diminishes the novelty of the authors' finding. I recommend repositioning the citation to clearly distinguish between the authors' original results and prior work. Please review other reference placement, especially in the results section.
5. Figure 2 caption – 2 MHz, not Mhz.
6. Pg 12, line 417 – This sentence is confusing: "Data acquisition. All applied signals were generated at 5MHz." Do you mean acquired?

Reviewer #4

(Remarks to the Author)

Reviewer #5

(Remarks to the Author)

I co-reviewed this manuscript with one of the reviewers who provided the listed reports. This is part of the Communications Engineering initiative to facilitate training in peer review and to provide appropriate recognition for Early Career Researchers who co-review manuscripts.

Reviewer #6

(Remarks to the Author)

I co-reviewed this manuscript with one of the reviewers who provided the listed reports. This is part of the Communications Engineering initiative to facilitate training in peer review and to provide appropriate recognition for Early Career Researchers who co-review manuscripts.

Version 1:

Reviewer comments:

Reviewer #1

(Remarks to the Author)

The authors' effort to address the previous review and better highlight the advantages and limitations of this method is appreciated. Despite suboptimal acoustic parameter selection, which limits the capabilities of this work to demonstrate spatial specificity using ultrasound and the lack of non-invasive EEG measurements, it has sufficient merit to be published. The final version is well-structured and well-written.

I have no further comments.

Reviewer #2

(Remarks to the Author)

The authors have addressed the concerns raised in my previous review in a thoughtful and comprehensive manner. In particular, the additional clarifications regarding potential auditory confounds and the distinction between pulsed and continuous ultrasound regimes help to contextualize the findings more clearly. The revisions to the figures, methodological descriptions, and inclusion of relevant references further strengthen the manuscript. While certain experimental limitations remain inherent to the approach, the authors have provided sufficient rationale and transparency to allow readers to interpret the results appropriately. Overall, I am satisfied with the revisions and consider the manuscript substantially improved.

Reviewer #3

(Remarks to the Author)

This was an extensive review. The authors did a thorough job addressing my comments. They improved clarity, contextualization, and discussion of limitations. I have no further comments.

Reviewer #4

(Remarks to the Author)

See attached PDF.

Reviewer #5

(Remarks to the Author)

I co-reviewed this manuscript with one of the reviewers who provided the listed reports. This is part of the Communications Engineering initiative to facilitate training in peer review and to provide appropriate recognition for Early Career Researchers who co-review manuscripts.

Reviewer #6

(Remarks to the Author)

I co-reviewed this manuscript with one of the reviewers who provided the listed reports. This is part of the Communications Engineering initiative to facilitate training in peer review and to provide appropriate recognition for Early Career Researchers who co-review manuscripts.

Reviewers' comments:

Dear reviewers,

We want to thank you for your insightful assessments and support of our manuscript.

Reviewer 1: “The authors present an interesting work that extends acoustoelectric measurements to in vivo neural recording. The topic of the paper is very relevant to the biomedical engineering community, and the use of ultrasound for enabling spatial selectivity and improved SNR in EEG measurements will be of interest and value to many scientists and engineers. In general, the paper is well-structured and well-written.

Reviewer 2: “This study presents a novel method for recording neural activity in vivo by leveraging the acoustoelectric heterodyne interaction. The authors demonstrate, for the first time, the capability to detect both evoked and spontaneous brain signals at the focal point of a focused ultrasound field through frequency mixing with endogenous electric fields.”

Reviewer 3: “The proposed approach could offer substantial advantages over EEG and MEG.”

Reviewer 4: “The novelty of this paper is that it is the first time this will be done in vivo in a rodent model with EEG-level signals. The authors present methods and tests to reduce artifacts and to verify that the signals recorded were created by acoustoelectric effect. The authors have made good efforts to try to validate their methods and demonstrate that they were generating and recovering biopotential-related acoustoelectric signals. They were also forward in disclosing their difficulties managing artifacts and noise problems in doing their measurements and explaining their methods for working through them.”

In the sections below, we respond (in blue), point by point, to your comments (in black).

Sincerely,

Jean Rintoul

Reviewer #1:

The authors present an interesting work that extends acoustoelectric measurements to in vivo neural recording. The topic of the paper is very relevant to the biomedical engineering community, and the use of ultrasound for enabling spatial selectivity and improved SNR in EEG measurements will be of interest and value to many scientists and engineers. In general, the paper is well-structured and well-written. Despite the acoustic parameter selection, which limits the capabilities of this work to demonstrate spatial specificity using ultrasound and the lack of non-invasive EEG measurements, it has sufficient merit to be published following the required corrections. However, the manuscript can significantly benefit from some clarifications regarding the methods used, as well as some editing (see comments and questions below).

General question: Throughout this work, you used particular ultrasound parameters of 0.5 MHz and 1 MPa. This parameter selection raises a couple of issues and, therefore, should be explained in the paper.

1. As mentioned in the discussion, this frequency limits your ability to demonstrate spatial

selectivity in mice. Did you consider using a higher frequency? If so, why was this specific frequency selected?

This paper aims to provide the first rigorous study of acoustoelectric heterodyning of neural activity using ultrasound. Once the proof of concept is established, future work will aim to demonstrate the spatial focality. 500kHz is the most commonly used frequency in transcranial FUS experiments¹, making our study comparable to other rodent tFUS experiments.

We have previously published work² showing acoustoelectric characterization measurements at this frequency and include in the supplement free-field phantom experiments using the ultrasound cone from the *in vivo* rodent experiments confirming acoustoelectric spatial specificity (See **Figure S3 f,g**).

2. Additionally, this relatively high pressure, combined with a low frequency, results in a relatively high MI (although still below imaging limits). Did you monitor for cavitation events or local temperature change?

In this article we measured the temperature change described in **Supplement S11** which confirms the thermal index was exceeded. We did not expect cavitation as the Mechanical Index (MI) was below the safety limits for diagnostic ultrasound imaging ($MI < 1.9@500kHz$)³, which are set below cavitation thresholds. Both temperature increase and the risk of cavitation are now addressed in the discussion section.

A few comments and questions:

3. Please provide a more detailed explanation of the result in Fig. 3 h. This result was initially not intuitive to me, and readers could benefit from an explicit explanation of why we expect the SNR with and without US to be the same in this context.

The SSVEP SNR (dB) in **Fig 3 h** was computed to determine if the neural evoked potential itself was significantly affected in by the application of ultrasound. For instance, if the application of continuous ultrasound significantly inhibited the endogenous signal, acoustoelectric neural recording would not be feasible. The text in the figure has been updated to specify it is the SSVEP SNR, and the article text clarified: 'The signal-to-noise ratio of the evoked SSVEP was then calculated for both the ultrasound and no ultrasound groups (**Fig 3 h**) (US group mean \pm s.d. = **20.91 \pm 14.55** ; No US group ;mean \pm s.d. = 19.05 \pm 7.73 $\# \ddagger 0.60$, $P = 0.54$), indicating that SSVEPs can be measured during continuous ultrasound stimulation and are not significantly attenuated by the application of ultrasound, enabling the acoustoelectric neural recording paradigm to be tested *in vivo*.'

4. How do you explain the temporal variability in the correlation presented in Figure 6C? It is not evident to me why the acoustic phase aberration due to the propagation of acoustic waves through heterogeneous media is time-dependent (line 287 in the discussion).

The discussion section has been expanded to include standing waves inducing periodic pressure variation which may induce amplitude changes in the heterodyned signal amplitude: "These inaccuracies could be due to acoustic phase aberration due to the acoustic waves' propagation through heterogeneous media which could be overcome by the implementation of phase aberration techniques³, standing waves induced by reflections creating periodic pressure changes, or non-coherent sampling caused by small timing differences between the signal generation and measurement hardware clocks⁴." It is also worth noting that the SNR depends on the neural signal amplitudes which fluctuated naturally over time, leading to SNR variance.

5. Figure 2a, please add the stereotaxic instrument to the illustration.

The stereotaxic instrument is photographed in **Supplemental Note 1b**.

6. Figure 2c, please rephrase this sentence to make it more readable.

Figure 2c legend has been rephrased to be more readable.

7. Line 432: Please add the model of the hydrophone.

The model number of the hydrophone has been added (Model Number: NH0200)

8. Supplementary information: Figure S2 instead of Figure S8 in line 50.

Done – Supplementary figure numbering is updated.

Reviewer #2:

This study presents a novel method for recording neural activity in vivo by leveraging the acoustoelectric heterodyne interaction. The authors demonstrate, for the first time, the capability to detect both evoked and spontaneous brain signals at the focal point of a focused ultrasound field through frequency mixing with endogenous electric fields. Using a 500 kHz continuous wave ultrasound and in vivo rodent models, the team records steady-state visual evoked potentials (SSVEPs) and successfully demodulates neural signals via both Hilbert envelope and in-phase/quadrature (IQ) techniques. The authors validate the acoustoelectric mechanism using two key control paradigms: frequency specificity and acoustic isolation tests. Their findings indicate that acoustoelectric neural recording enables spatially resolved, artifact-resistant electrophysiological measurements even during continuous ultrasound exposure, offering a potential path toward non-invasive, high-resolution neural recording techniques surpassing traditional EEG.

While the authors employ 1-second onset and offset ramping—a known technique to reduce auditory transients—the manuscript does not explicitly address the possibility of auditory confounding effects, particularly under the 80 Hz PRF condition used in some experiments. Rodents are known to exhibit neural responses to low-frequency ultrasound pulses via auditory pathways, raising the possibility that the recorded signals may partially reflect indirect auditory activation. Additionally, although the primary aim of the study is signal readout rather than neuromodulation, the use of 1 MPa continuous-wave ultrasound raises the possibility of direct neuromodulatory effects, especially in the spontaneous activity recordings. Without controlling for such effects, it remains unclear whether the recovered signals truly reflect pre-existing endogenous neural activity or ultrasound-induced changes. A more thorough discussion or additional controls (e.g., auditory masking, sham stimulation, or cortical monitoring) would help strengthen the attribution of the recorded signals specifically to acoustoelectric demodulation.

Auditory confounds have been reported at the pulse repetition frequency (PRF) when the PRF occurs within the auditory range⁵. The 80Hz PRF experiments were done to explore previous work attempting acoustoelectric neural recording and our results were artefactually confounded by the presence of spike harmonics (see **Supplementary Section 8**). These results could also be artefactually confounded by the auditory confound as the reviewer has pointed out, which is now added into the main manuscript. The two artefact tests outlined in the article - the frequency specificity of the modulation products and the acoustic isolation test can rule out both the auditory

confound and spike harmonics, and we show these tests successfully using averaged SSVEPs and continuous ultrasound (**Fig 4,5**), and spontaneous neural activity (**Fig 6**).

With regards to direct modulatory effects from ultrasound affecting the SSVEP, this is possible and explored in **Figure 3** to show that application of a continuous acoustic wave does not significantly inhibit the SSVEP, enabling SSVEP measurement with concurrent ultrasound.

Although the text confirms the use of intracranially implanted electrodes for signal acquisition, Fig. 1 labels the measurement electrode as "external," which could be misleading. For clarity, it may be helpful to revise the label or caption to indicate that the electrode tip is intracortical while only the connector is external.

The concept picture has been updated to remove the word external and depict the electrode connector more accurately.

While the authors acknowledge the emergence of standing waves in the mouse model (**Fig. S7**), it remains unclear how these acoustic field distortions may have impacted the spatial specificity and fidelity of the recovered acoustoelectric signals. A more detailed discussion—or ideally, a sensitivity analysis—would help assess whether the measured heterodyne signals can be reliably attributed to the intended focal region under these acoustic conditions.

Supplemental Note 10 shows standing wave reflections occur when the acoustic wavelength is in the same order of magnitude as the mouse brain. Under these conditions, any acoustic signal inside the mouse head is likely to pass through the visual cortex which forms 12.2% of the entire cortical area⁶ in a mouse. The discussion section outlines standing waves may be a cause of inaccuracy in the recovered signal. Though we can simulate the standing wave reflections in a standard mouse finite element model, this does not accurately reflect the real *in vivo* scenario, which varies based on small angular and material differences in the mouse anatomy, making an accurate sensitivity analysis beyond the scope of this article. The discussion section suggests that future studies could use either a higher frequency (shorter wavelength) transducer or a larger mammal to reduce acoustic standing waves.

To address the simultaneous measurement of acoustoelectric heterodyning and spatial specificity, we have results in **Supplemental Note 3**, showing the free field phantom which has fewer reflections and standing wave issues, confirming both acoustoelectric heterodyning and spatial specificity simultaneously.

Although conceptually distinct, Yu et al. (2016, IEEE T-BME) reported noninvasive electrophysiological imaging of tFUS-induced brain activity using scalp EEG and source localization. Citing this work would help contextualize the current study within the broader spectrum of electrophysiological approaches to tFUS monitoring.

This article has now been cited to contextualize the section on measuring electrical signals with concurrent continuous ultrasound with this article listed in the references:

K. Yu, A. Sohrabpour and B. He, "Electrophysiological Source Imaging of Brain Networks Perturbed by Low-Intensity Transcranial Focused Ultrasound," in IEEE Transactions on Biomedical Engineering, vol. 63, no. 9, pp. 1787-1794, Sept. 2016, doi: 10.1109/TBME.2016.2591924.

Reviewer #3:

This manuscript introduces and validates a method for recording in vivo neural signals through

acoustoelectric heterodyning. The technique uses focused ultrasound to modulate endogenous electrical signals from the brain, allowing spatially specific electrical recording deep in the brain. The proposed approach could offer substantial advantages over EEG and MEG but some aspects require further clarification.

Major Points

1- The manuscript does not address a critical potential confound that the ultrasound itself, especially at 1 MPa continuous wave, may modulate neural activity. Both thermal and mechanical effects of ultrasound are known to influence excitability, synaptic transmission, and oscillatory dynamics. This raises

1a. safety concerns and

1b. the possibility that the recorded signals are not purely passive reflections of endogenous brain activity but may be altered by the recording method itself. I recommend the authors cite relevant LIFU neuromodulation studies, clarify whether any steps were taken to assess or limit neuromodulatory effects in the present setup (e.g., short durations, behavioral baselines, histology).

1a. Firstly, your comments on safety are appreciated and further studies would need to be performed to show safety and histology results with further waveform optimization, as outlined in the discussion, lending translation to humans as a sensible next step for future studies. This article focused only on feasibility of the technique to detect neural activity in mice with artefact test exploration to prevent artefactual confounds.

1b. **Figure 3** addresses measuring SSVEPs with and without continuous ultrasound (as a control for the SSVEP measurement), showing no significant between continuous ultrasound and no ultrasound groups SSVEP SNR. From a safety perspective, the mechanical index (MI) is kept below 1.9 which is within the safety guidelines for diagnostic ultrasound imaging. The Thermal Index does surpass the safety limitations, and **Supplemental Note 11** addresses temperature change measurements in a free-field phantom.

The discussion was expanded to include reference to the safety guidelines from the ITRUSST consortium in further work: 'Key factors for this technique's successful translation through to humans will be detecting the small acoustoelectric signal outside the human skull, while being adherent to the recommended safety guidelines⁷. In this article we focused on the feasibility of acoustoelectric neural recording with associated artefact tests in a rodent model, which lent itself to using a long continuous acoustic wave to simplify the signal recovery process. Although the mechanical index (MI) was within the safety guidelines for diagnostic imaging, the temperature change (**Supplementary Note 11**) was outside of the thermal index safety guidelines. To enable translation of this technique to humans, further acoustic waveform optimization would be required to optimize SNR of the modulated waveform through the skull, while adhering to both the MI and TI safety guidelines. '

1c. Pg. 4 line 147 - The manuscript states that the carrier-frequency and DC offset signals are only present during ultrasound trials. In order to distinguish whether signals are the result of passive detection or ultrasound-induced neuromodulation of neural activity, perhaps apply the technique to a non-evoked state (e.g., in anesthetized or postmortem tissue) to determine if the signal still appears, confirming it's not neuromodulatory in origin.

Supplementary Note 7 has been added, where the DC offset trials were repeated in saline with the same result, and a sentence added to the main manuscript indicating the finding: 'The DC

offset remained present in a similar set of continuous ultrasound saline phantom trials, indicating the DC offset was not due to neural activity (See **Supplemental Note 7**).'

2- The ultrasound frequency used 500 kHz produces a focal region several millimeters wide, comparable to the entire size of a mouse brain. How can the method resolve or isolate neural activity from localized sources if the ultrasound interacts with multiple brain regions simultaneously? Clarify in the discussion how higher frequencies or larger animal models could solve the focality issue while overcoming attenuation of FUS and what steps are needed toward human translation.

We have expanded the limitations and steps toward human translation paragraph in the discussion:

'Limitations in the experiments in this article are that we could not show focal specificity within the mouse model using a large wavelength($\approx 3\text{mm}$) 500kHz ultrasound transducer, due to emergent standing waves which occur when the wavelength is a sub-multiple of the mouse head diameter ($\approx 15\text{mm}$) with finite element simulation shown in **Supplementary Note 10**. Future work using a larger mammal or a smaller wavelength transducer may be able to overcome this limitation.'

To steps to human translation:

'Key factors for this technique's successful translation through to humans will be detecting the small acoustoelectric signal outside the human skull, while being adherent to the recommended safety guidelines⁷. In this article we focused on the feasibility of acoustoelectric neural recording with associated artefact tests in a rodent model, which lent itself to using a long continuous acoustic wave to simplify the signal recovery process. Although the mechanical index (MI) was within the safety guidelines for diagnostic imaging, the temperature change (**Supplementary Note 11**) was outside of the thermal index safety guidelines. To enable translation of this technique to humans, further acoustic waveform optimization would be required to optimize SNR of the modulated waveform through the skull, while adhering to both the MI and TI safety guidelines.'

3- The manuscript does not discuss how the implanted brain electrodes may interact with the ultrasound field. This is an important omission, as the presence of an electrode in the acoustic focus can introduce potential confounds, such as viscous heating artifact at the metal-tissue interface, acoustic scattering or reflection, micro-vibrations of the electrode, etc.

Supplementary Note 4 has been added exploring the possible vibration artefact confound. Prior free field phantom experiments showed the acoustoelectric amplitude dependence on the salinity of the medium when the electrodes remained in the same position, suggesting vibration is not the cause of the acoustoelectric effect. Furthermore, there is an angular dependence between the electric and acoustic field, using the same electrodes in each test, which is in accordance with the acoustoelectric equation (**S4 c-h**). Hence, the measured spatial map differences cannot be explained by electrode vibration alone. Furthermore, prior work at the University of Arizona on cardiac activation mapping⁸, where the measurement electrode was kept in the same position and the ultrasound was moved also suggest that the effect is not due to vibrations at the electrode-tissue interface, as this would not induce a variant cardiac signal spatial map as only the acoustic source moved.

4- IQ demodulation assumes synchronization between the incoming signal and the reference oscillator (carrier). At 500 kHz, even a microsecond-scale delay may introduce significant phase error. This can potentially distort the I and Q components, especially when trying to track very

slow (<20 Hz) modulations like neural signals. In addition, drifts in ultrasound amplitude or phase due to tissue inhomogeneity, motion, or coupling instability may occur. Please discuss whether these factors may limit the generalizability or robustness of the method in less controlled settings (e.g., freely moving animals or chronic implants).

IQ demodulation was used as it does not require synchronization between the incoming signal and the reference oscillator. The reference carrier signal in IQ demodulation does not need to be in phase for demodulation to work as it employs I and Q components which are 90 degrees shifted from each other so that the sum always adds to 1. Further information on IQ demodulation is available in the prior physics demonstration article². The specific computation for IQ demodulation in this article is shown in the Github repository, with associated Figshare data file supplied with the article here: <https://figshare.com/s/2ca17d7ca823aaa59748>

5- Pg. 14 - The manuscript refers to the mechanical index (MI) to describe ultrasound safety and exposure, but MI alone is insufficient to characterize the acoustic field. Please include acoustic intensity (Ispta, Isppa), thermal index and other relevant parameters.

Since we used a continuous acoustic wave, the safety parameters used to average the usual ultrasound pulsing schemes (ISPTA and ISPPA) were not as applicable, and instead we used the core safety indices of mechanical index and thermal index. In standard ultrasound pulsing patterns, ISPTA (spatial peak temporal average intensity) is the average intensity over time, while ISPPA (spatial peak pulse average intensity) is the intensity of a single pulse. Since we use a continuous wave our ISPPA is the same as the ISPTA. We can conservatively derate this calculation for the tissue properties of white matter at the focus of the ultrasound, leading to an ISPPA estimate of 10.01W/cm², which is within the FDA guidelines⁹ for cephalic ultrasound which suggest a maximum safety value range of ISPPA < 190.01W/cm².

6 - Emphasize more clearly in the abstract and introduction how this work advances beyond prior UCSDI methods and recent acoustoelectric studies. Quantitative results in the abstract should also strengthen the work importance.

This sentence is added to the abstract:

'Recent discoveries that the acoustoelectric interaction involves frequency mixing enable improvements on prior reports of acoustoelectric imaging in cardiac tissue, so that neural signals can be recovered in an in vivo rodent model'.

7- Discuss potential sources of residual error in the demodulated signals, e.g., phase aberration, as mentioned briefly in the discussion, and whether phase correction techniques might mitigate this.

The discussion section has been updated, to more clearly describe that the residual error may be improved with the implementation of phase aberration techniques: 'These inaccuracies could be due to acoustic phase aberration due to the acoustic waves' propagation through heterogeneous media which could be overcome by the implementation of phase aberration techniques³, or non-coherent sampling caused by small timing differences between the signal generation and measurement hardware clocks⁴.'

8- Not all frequency choices are well justified, for example, in pg. 3 line 107, 80 Hz PRF, Hilbert enveloped +/-50Hz. A brief explanation should be added for why each frequency was chosen, what prior literature (if any) supports this, and whether different frequencies were tested or ruled out.

The experiments presented in the main text of the article all use continuous wave and successfully pass both the frequency specificity test and acoustic isolation test.

The 80Hz PRF experiments were performed to explore the same parameters used in prior work attempting acoustoelectric neural recording that we found to be artefactually confounded by the presence of spike harmonics which are explored in detail in **Supplementary Section 8**. These results could ALSO be artefactually confounded by the auditory confound, which is now added into the main manuscript - 'This frequency specificity test determines the mechanism behind the modulation is due to heterodyning, with negative *in vivo* examples evidenced where there is periodic broadband noise caused by VEP harmonics inducing an artefactual demodulation in the frequencies below 1kHz (**Supplementary Note 8**). Furthermore, demodulation at the 80Hz PRF could also be susceptible to the auditory confound⁵, suggesting that any acoustoelectric neural recording previously reported at low PRF's is artefactually confounded unless appropriate tests are in place.'

Minor comments

1. Add brief comparative discussion of how this method fits among existing high-resolution neural recording techniques (e.g., high-density ECoG, optogenetic reporters, MEG).

Done. The following has been added to the discussion:

'Since ultrasound enables focusing via spatial interference at length scales relevant to the human brain, acoustoelectric neural recording has the potential to be focused to small regions both at cortical layers and at depth¹⁰. Unlike high-density ECoG¹¹, acoustoelectric neural recording has the potential to be non-invasive which reduces the risks incurred by surgery¹², and does not require transgenic modification of optogenetic reporters¹³ enabling human translation to be considered in future work. In contrast, the BOLD signal in fMRI capture the slow metabolic responses to neural activity instead of direct measures of the electrical neural activity. Acoustoelectric neural recording has the potential to enable previously inaccessible insights into how we understand the brain and diagnose brain disorders such as epilepsy¹⁴, obsessive compulsive disorder¹⁵ (OCD) and depression¹⁶.

Potential advantages of acoustoelectric neural recording are that acoustoelectrically detected signals are separated from the neighboring neurons as only the acoustic focal volume is modulated up to this higher frequency, enabling far greater spatial specificity in recording. Furthermore, acoustoelectric neural recording has a different noise profile to EEG or MEG, with potential advantage that the thermal noise floor^{17,18} is much lower around the acoustic frequency compared to the original low frequency endogenous signals, giving it signal transmission advantages^{19,20}. Compared to EEG, the detection of the small acoustoelectric neural signal need not be limited to electrode position and size as the spatial locus is controlled by ultrasound, enabling larger electrodes to be used to average away thermal noise^{17,21}.

2. Provide a conceptual roadmap toward non-invasive application in humans. This would highlight key technical gaps (e.g., deeper penetration, skull aberration correction, safe intensity limits).

Done – updated in discussion:

'Key factors for this technique's successful translation through to humans will be detecting the small acoustoelectric signal outside the human skull, while being adherent to the recommended safety guidelines⁷. In this article we focused on the feasibility of acoustoelectric neural recording

with associated artefact tests in a rodent model, which lent itself to using a long continuous acoustic wave to simplify the signal recovery process. Although the mechanical index (MI) was within the safety guidelines for diagnostic imaging reducing the likelihood of cavitation, the temperature change was outside of the safety guidelines (**Supplementary Note 11**). To enable translation of this technique to humans, further acoustic waveform optimization would be required to optimize SNR of the modulated waveform through the skull, while adhering to the safety guidelines.'

3. Supplementary Figure S2 wrongly numbered.

This has been corrected.

4. Pg. 2 line 93 – Reference 23 seems misplaced. As written, the placement of the citation suggests that this result was reported by Reference 23, which diminishes the novelty of the authors' finding. I recommend repositioning the citation to clearly distinguish between the authors' original results and prior work. Please review other reference placement, especially in the results section.

All references have been checked.

5. Figure 2 caption – 2 MHz, not Mhz.

Fixed

6. Pg 12, line 417 – This sentence is confusing: "Data acquisition. All applied signals were generated at 5MHz." Do you mean acquired?

The signals were generated at 5MHz and acquired at 2MHz.

Reviewer #4: (transcribed from email pdf attachment).

Summary

This study sets out to demonstrate the use of focused ultrasound to heterodyne and record neural activity signals at the brain surface. This phenomenon, known as the acoustoelectric effect, has been used in previous studies for imaging electrical dynamics, most notably by Witte and colleagues at the University of Arizona. The novelty of this paper is that it is the first time this will be done in vivo in a rodent model with EEG-level signals.

The authors present methods and tests to reduce artifacts and to verify that the signals recorded were created by acoustoelectric effect. They also present a novel setup in which they simultaneously record both evoked potentials and the acoustoelectric heterodyned signals through the same brain surface electrodes using a single wideband amplifier and data acquisition system. Finally, they compare the direct brain surface recordings to the signals recovered by demodulating the high-frequency acoustoelectric signals.

The direct evoked potential recordings and demodulated acoustoelectric signals presented appear compellingly similar, and this supports the case that they successfully acquired the same underlying signal through these two different modalities. However, artifacts, variability, noise, and other limitations in their data make it difficult to be sure that the data show acoustoelectric capture of microvolt signals and not some form of crosstalk in their

setup. More validation is needed to establish this. The paper also makes claims about potential capabilities for focal recording of EEG signals with remote or surface electrodes and focused ultrasound, but these claims are highly speculative and not supported by the current work.

Major comments (general and by line number)

- Although the paper is well referenced, the phrasing and argumentation in many places are not consistent with traditional understandings of neuroscience and electrophysiology. For example, Hodgkin-Huxley is a model for describing membrane-level dynamics, not brain level dynamics as stated by the authors. Also, while it is true that EEG is spatially limited by attenuation from the skull, the bigger challenge for higher resolution neural recording is that the signals become more localized as smaller populations of neurons are targeted. Even without the skull, the volume conduction of the head and scalp makes spatially specific neural recordings difficult without using smaller electrodes placed close to the neural tissue of interest. Additional editing by a subject expert in electrophysiology and/or neurophysiology would likely improve reception of the paper by those audiences.

We have removed any mention of the Hodgkin-Huxley model, replacing it with the following:

'There are currently no non-invasive techniques which can detect neural electrical activity in the brain with high spatial specificity and depth. Despite the importance of measuring electrical brain activity for understanding brain disorders and function, current non-invasive neuronal recording techniques such as electroencephalography²² (EEG) are limited to diffuse cortical measurements of synchronized, large-scale events, where the electrical neural signals are strongly attenuated by the skull²³. Magnetoencephalography²⁴ (MEG) can provide high spatial resolution measurements of electrical neural activity, but only at the cortical layers. Blood-Oxygenation-Level-Dependent (BOLD) functional magnetic resonance imaging²⁵ (fMRI) has three-dimensional spatial resolution but it is only an indirect and slow measure of changes in neural electrical activity.'

- The proposed use of acoustoelectric heterodyning to frequency shift a small focal area of neural signals is very interesting and an exciting possibility. However, as with EEG-level neural signals themselves, the very small frequency-shifted signals created at the ultrasound focus are also likely to prove difficult to detect at farther points across the volume conductor of the head and through the skull. The authors imply that this is what they are doing in Figure 1 and claim in their discussion, but all the test setups and their ephys experiments work by placing the acoustoelectric detection electrode directly in the ultrasound focus, where the acoustoelectric signal is going to be at its maximum. Moreover, the holes created in the skull where the electrodes are placed may also be concentrating the VEP electrical gradient around the electrode, further enhancing a potential acoustoelectric signal. To make the claims that microvolt, biopotential-related acoustoelectric signals are detectable remotely or through the skull, the authors should at least provide some sort of simulation model to predict that this is achievable within the noise limits of electrodes and electronic amplifiers. A convincing bench model with recording electrodes outside of the ultrasound focus would be even better. It's not possible to claim that remote sensing is feasible based on the tests described in Figure S8.

The concept **Figure 1** has been updated to reflect the *in vivo* electrophysiology experiments in the manuscript.

There are free field phantom experiments in **Supplementary Note 3 and 4** showing both the heterodyning effect and the focality of the acoustoelectric effect, where reflections were

minimized by using a larger 20cmx20cmx50cm tank and the use of acoustic damping material. These experiments are discussed in further detail in the previously published Nature Communications Physics article².

To verify that we achieved similar results to the free-field phantom, we used a saline petri dish bench model within the same space available to the mouse within the *in vivo* electrophysiology instrumentation. **Figure 2 b** confirms the focality of the sum and difference frequencies by manually moving the stereotaxic instrument over a 1cm square in 0.5mm increments. At no point do we claim that remote sensing is feasible in **Figure S8**, addressing the reviewers concerns.

- The authors have made good efforts to try to validate their methods and demonstrate that they were generating and recovering biopotential-related acoustoelectric signals. They were also forward in disclosing their difficulties managing artifacts and noise problems in doing their measurements and explaining their methods for working through them. However, the combined nature of the signal path, the complexity of the signal processing, and some of the data (such as the acoustic signal inducing DC shifts in the electrode recordings) still make it difficult to rule out that some other type of signal mixing is present in their setup and that the observed modulation is not truly based on acoustoelectric effects. In addition, all the recordings seem to have been made with the recording electrodes in the ultrasound field, raising the possibility of ultrasound interactions with the electrodes themselves. More bench validation of the analog signal processing chain and verification of the digital signal processing with simulated data and noise signals are needed to establish that the claimed acoustoelectric modulation and detection are correct.

A supplementary note on the DC shift has been added into as **Supplement Note 7**, showing this is not generated by neurons. The bench validation of the signal processing chain was done in the previously published Nature Communications Physics article², which has a supplemental section validating signal generation and recording. This is the same system used in the *in vivo* electrophysiology experiments. In this article we have shown characterization of the equipment in **S1,S2,S3** and **S9**, and tried to describe all instruments in detail in **Methods**.

72. More information should be provided regarding the size of the Pt/Ir electrodes and the geometry of their placement (distance apart?). Without that information, it's impossible to know the applied electric field gradient or reproduce the setup. In addition, in Figures 2a and S8b, it appears that the electric field is applied orthogonal to the ultrasound propagation direction, but in Figure S8c, it appears that the electric field is provided in parallel with the ultrasound beam. Moreover, the electrical setup used to apply this signal is relevant and should be described. Ideally, the VE would be applied via an isolation transformer to create a floating voltage source in the saline.

Supplemental Note 2 has been added to describe the electrodes in more detail, and the main article updated to include electrode spacing information:

'Mice underwent a recovery surgery to enable the recording of visual evoked potentials (VEPs) with Platinum Iridium electrodes placed in the visual and motor cortex (see **Methods** for surgery details), 7mm apart from each other. '

Fig 2 Legend update to include electrode spacing information:

'**c**, Acoustoelectric saline phantom comparisons using a rectangular versus Kaiser window. Two platinum-iridium electrodes measure the electric field in solution (μm), spaced 7mm apart,

sampling rate 5MHz and preamplifier gain at 5000, 12 second duration, pressure output 1MPa at 500kHz in all recordings and electric sinusoid () at 10Hz 35

In **Fig 2 b**, is applied by an isolation transformer according to the set up described in the supplementary section in the prior article². For very low frequency signals (i.e. **Fig 2C**), we could not use the isolation transformer as its bandwidth was 5kHz-2MHz so we used the signal generator directly (**Fig 2 c-e**). The isolated differential measurement using the SR560 preamplifier, meant that common mode signals were removed.

73. The configuration of the recording electrodes for the acoustoelectric signals is not really described other than to say that they are Pt/Ir electrodes. Their size (impedance?) and arrangement should be described more clearly. In the supplemental material, it appears that the acoustoelectric detection electrode and the VE electrodes are moved together on an assembly, but this is not described.

There is a new supplemental section on the electrode configuration and their impedances in both saline and a mouse **Supplementary Note 2**, which is also referenced on line 73.

We have added a statement in **Supplementary Note 3** – acoustoelectric and cone characterization, which references the full description of the XYZ free-field phantom stage used to characterize equipment: ‘For further characterization and description of the phantom XYZ stage please see the Methods and Supplemental material previously reported². ‘

77. The “non-focal electric field” or electrical artifact from the transducer does appear to be “non-focal”, but it also appears to vary significantly with position and to have structure in Figure 2(b)iii. This is very strange and implies that something else is possibly going on besides electrical coupling. For example, is it possible that the ultrasound is somehow generating an electromechanical artifact at the charge double layer at the electrode-saline interface? Or causing capacitive microphonic signals at the insulation of the wires? If there were other carrier transduction mechanisms present in the test setup, it could help explain some of the variability seen in 2(b)iii.

At line 77, we have added a reference to **Supplementary Note 4** which investigates dependence on acoustic vibration at the electrodes, suggesting the medium is the source of the acoustoelectric signals and not vibration at the electrodes. Electrical only frequency mixing is investigated later in **Figure 5**, where in vivo evidence suggests the sum and difference frequencies are not due to electric only mixing.

The slight unevenness trend noted by the reviewer in **Figure 2 (b) iii**, is likely due to their being a small angular difference between the saline petri dish and the end of the water filled transducer cone which meant the capacitively coupled electrical artefact was not on a single plane, hence there is a reduction in amplitude with increased distance from the end of the capacitively coupled ultrasound.

81. Here again, the details of the excitation and recording electrodes are important to understanding the work and should be further described. The acoustoelectric signal is proportional to the gradient of the electric field (and the ionic current) where the ultrasound is focused. Without knowing the electrode geometry and spacing, it is impossible to know the geometry of the induced gradients and if the applied fields are good models for the electric fields and currents present in biological tissue.

Supplementary Note 2 has been added with details on the recording electrodes in both the saline and mouse arrangement, with their relative impedance with respect to frequency when positioned 7mm apart.

109. It's not clear if the authors did a Pearson correlation that was shifted in phase to find the maximum correlation point. A simple direct Pearson correlation measure is not a good index of fidelity or similarity for periodic time-domain signals. For example, for a sine wave, if the recorded signal is identical but 90 degrees out of phase with the reference signal, the correlation would be zero. A more conventional metric like THD and/or some kind of characterization of the added noise in the reconstructed signal would be more appropriate here.

The reference carrier signal in IQ demodulation does not need to be in phase for demodulation to work as it employs I and Q components which are 90 degrees shifted from each other so that the sum always adds to 1. This is the key advantage of IQ demodulation over other demodulation schemes, making the Pearson correlation metric directly applicable without need for phase shifting. The specific computation for IQ demodulation described in detail in **Methods**, with code in the Github repository, and an associated Figshare data file supplied with the article located here: <https://figshare.com/s/2ca17d7ca823aaa59748>. Further, our recent previously published article shows the IQ demodulation algorithm in further detail with relevant references for more detailed overview²

144. The DC artifact created by the ultrasound is very problematic as it implies that the 500kHz artifact is somehow being rectified or demodulated through a nonlinearity in the signal path. As a result, any amplitude modulation of the recorded 500kHz carrier (even acoustoelectric signals) would likely be demodulated to baseband signals that are added to the recording. A simple test of this would be to slightly amplitude modulate the source for the 500kHz carrier at 10Hz and see if a 10Hz signal appears in the low-pass filtered signal. This should be investigated further as this artifact mechanism could impact the fundamental claims of the work.

Supplemental Note 7 has been added to show the DC offset induced by the ultrasound is not induced by neurons. The front-end filters on the SR560 preamplifier remove this artefact to keep the signal below saturation with raw signals confirming this shown in **Figure 3 b**, and there is no evidence that the DC offset is induced through rectification.

The reviewer suggests a two-tone electrical test with two electrical signals at the same frequencies as seen in vivo (500kHz and 10Hz) using the same preamplifier settings, which is presented in **Supplementary Note 9**.

The results section **Evidence frequency mixing is due to the acoustoelectric effect**, addresses electrical only mixing, with multiple tests suggesting the dominant cause of the sum and difference frequencies is not due to electrical only mixing.

208. The use of the air gap is a clever approach for verification of mechanism, but the final SNR metric used here is unconvincing and difficult to judge without knowing the variability of the background noise itself. It could be that the air gap simply adds lots of noise in some cases. Moreover, the images in Figures 5(b)iii and 5(c)iii show a clear, compelling example of the difference in mixing between connected vs. isolated cases, but the SNR values in Figure 5(f) appear to vary wildly by orders of magnitude.

It also doesn't seem reasonable that the proposed SNR method can give reliable SNRs as low as -20 dB. Is it possible to apply a better SNR metric here? Or explain why so much of the SNR data don't appear to demonstrate the effect as well as the examples in 5(b)iii and 5(c)iii.

The signal to noise ratio is low due to the electric artefact (described in **Supplemental Note 5**) also being present around 500kHz. There is a large variance in the signal to noise because of the highly variant noise floor which can be seen in **Figure 2 c ii** and again in **Figure S6 b iii**, as well as a naturally occurring variation in the VEP amplitude over the course of the experiment and between mice.

The noise floor in the signal to noise ratio (dB) is calculated as per the description in the **Figure 5** legend below:

'f, Acoustic isolation test comparing signal-to-noise ratio (dB) between acoustically connected and isolated groups with SNR (dB) calculated as mean of $(\Delta + \Sigma) / 2$, and noise as the mean of the spectral bins ± 2 around the Δ frequency'. An SNR < 0dB, indicates that noise dominated the signal.

By repeating the SNR measure we can interpret where the difference is significant which is the key information reported, despite there being both VEP amplitude and electrical artefact variation.

230. This paragraph shows effort by the authors for trying to eliminate possible non-acoustoelectric sources of modulation. However, the claims here are strongly in conflict with Figures 3(b) and 3(c) given the strong DC artifact signal created in the electrode recordings when the 500kHz ultrasound carrier is present. This is generally indicative of some form of nonlinear rectification of the AC interference into a DC signal. If the AC signal has amplitude modulation, this translates into the demodulated signal getting mixed into the signal path. Nonlinear interactions between the amplifier and the electrode electrochemical interface can sometimes be a source of these effects. This should be analyzed further and/or explained.

With regards to the DC artifact in **Figure 3 b and 3 c**, we do not assume this is due electric mixing only or caused by rectification but observe only that it exists and front-end filtering can reduce it to enable higher gain on the measurement preamplifier without nearing saturation.

The reviewer points to the risk of electric only mixing occurring due to the electrode electrochemical interface. To explore electric only mixing, we have tests described in the **Figure 5** results section '**Evidence frequency mixing is due to the acoustoelectric effect**'. Here, we can showcase **Figure 2b** which shows focality of the sum and difference frequencies with the acoustic field, the acoustic isolation test, and the two-tone electrical test shown in **Supplementary Note 9**.

291. There are certainly noise and electrode impedance benefits to working with signals in the 500kHz range as opposed to 10Hz in biological tissue. However, the experiment in Figure S8 and the claim that high-frequency signals have improved transmission characteristics in saline (and the brain) suggest misunderstandings of electrode impedances and the propagation of voltages and currents in volume conductors. The results shown in Figure S8 are more likely explained by the higher impedance of partially polarizable electrodes like Pt and Pt/Ir at lower frequencies due to their

predominantly capacitive interface impedance. When the stimulation electrodes are subjected to a 1 mV AC voltage source, much more current is driven into the tissue at 500kHz due to the lower impedance of the electrodes at 500kHz. This in turn leads to higher voltages being measured by the measurement electrodes. The observations here are likely due entirely to the impedance of the stimulation electrodes rather than anything related to the propagation of electrical signals through tissue itself at these frequencies.

Agreed there are noise and electrode benefits to working with signals in 500kHz range, though these benefits also exist in the medium as well. The previous **Figure S8** has been removed and replaced with references to finite element simulation of electric fields which can exclude electrochemical electrode interface effects and show the same trends in the medium, using only Maxwell's equations. For instance, in this simulation article the electrode interface is excluded from the computation²⁰ – yet we still see that there are different impedances in the medium at different frequencies, changing the relative attenuation through the medium which is the key point we wished to highlight in the list of advantages and limitations. Further in vivo articles also report different conductivities and hence attenuation at different frequencies¹⁹. The IT'IS database on tissue properties also shows extensive testing reporting differences in conductivity at different frequencies below 1MHz (<https://itis.swiss/virtual-population/tissue-properties/database/low-frequency-conductivity/>).

The reviewer points out the importance of eliminating electric only mixing as a potential confound through their comments on electrochemical interactions. To address this, we show an invitro test in **Figure 2b** to differentiate between electric field only and acoustoelectric mixing **Figure 5** acoustic isolation test, and **Supplement 9** two-tone electrical mixing test. These form an argument against the measurements being due to electric only based mixing such as electrochemical effects at the electrodes.

294. The possibility that the acoustoelectric signal might be detectable by electrodes in other positions than the ultrasound focus area is interesting and worth investigating, but it is not supported by the current work and the erroneous conclusions drawn from Figure S8. Claims of advantages over state-of-the-art methods like EEG should be tempered.

The previous **Figure S8** has been removed from the manuscript. The discussion has been updated to clearly state the potential advantages and limitations, as well as a path to human translation.

Minor comments (general and detailed by line number)

- There are editing-level mistakes and word omissions throughout the manuscript. The inclusion of high numbers of significant digits in the figure captions should be reviewed. The descriptions of the methods are distributed in the Results section, the Methods section, the figure captions, and the supplemental material in ways that are difficult and tedious to follow. These could be revised for better flow and clarity.

All significant digits have been reviewed.

9. This is simply not correct. There are a wide variety of tools available for recording neural activity in deep brain areas with high spatial specificity used in both research and clinical practice (depth electrodes, sEEG electrodes, etc.). It seems the authors are trying to say instead that they would like to find non-invasive ways of doing focal

recordings in deep brain areas.

Thank you. The abstract has been updated to specify the non-invasive detection of neural electrical activity.

43. Why was citation 16 experimentation limited by the understanding of the underlying physics? Mention of literature by Witte is referenced, but there is a more recent paper by the group from 2020 (Alvarez et al., Applied Optics, vol 59, iss 36, pp 11292-11300).

Great - the citation has been changed to include the more recent paper, and we removed the sentence 'limitations to our understanding of the underlying physics'.

76. The parenthetical expressions for "sum" and "difference" appear swapped.

Corrected.

80. In Supplementary Note 2, Figure S8 g, h, and i, the amplitudes of the two side band signals (Δ and Σ) are significantly different, even though it seems the acoustoelectric mechanism should create them identically. This implies that the amplifier passband might not be flat and/or some sort of error might be present in the signal processing. This should be discussed, especially if it could affect the demodulation process.

This is a good observation. It is true that the passband of the preamplifier may not be completely flat, as the pass-band characteristics are only reported for resistive loads, and the measurements made in **Supplemental Note 3 g,h,i** involve complex loads which may induce a different response. An additional explanation is that the ionic mobility dependence reported by Lavandier and Jossinet^{26,27}, is frequency dependent as the conductance of solution is proportional to the frequency. Both these factors could easily lead to differences in the sum and difference frequency amplitudes.

82. Although the authors describe their differential electrodes for recording, they never mention the grounding configuration in their saline bench or animal experiments. Although differential amplifiers measure the voltages between two inputs, the saline and animals in these experiments must somehow be held within the common mode input range of the amplifier. Without a third grounding electrode for this, the subject is only weakly grounded to the amplifier through its input impedances, and recordings become very sensitive to common-mode (CM) noise sources and distortion as the recording subject floats around the CM input range of the amplifier or comes close to saturating the input amplifier stages. Approaching input saturation limits in the amplifier can also lead nonlinearities in the signal path.

The measurement system is floating from ground at both the electrode location and the preamplifier stage, to enable maximum common mode rejection between the two floating differential measurement probes. The head stage used has a floating preamplification design which can be found here: <https://www.thinksrs.com/products/sr560.html>), which differs from the more commonly used multi-electrode ECoG or EEG arrays.

The reviewer is referring to a grounded amplification system which is a common configuration used in electrophysiology where there are more than two electrodes. When more than two electrodes are used (i.e. EEG or ECoG arrays), the grounded amplification and referencing

scheme is required. Since we have only two electrodes, we can use the floating and isolated from ground differential system to maximize common mode rejection.

To avoid non-linearities in the signal path, we performed a series of tests isolating electric only mixing from acoustoelectric mixing i.e. **Figure 2b** to differentiate between electric field only and acoustoelectric mixing, **Figure 5** acoustic isolation test, and **Supplemental Note 9** two-tone electrical mixing test.

135. The multi-unit and local field potential recordings shown in Figure S4(c) show that their setup is capable of recording good VEP responses.

Thanks.

138. For a 2 Msps sample rate, a first- or second-order 1MHz low-pass filter is insufficient to prevent Nyquist-related aliasing. The filter will only have 3 to 6 dB of rejection at the corner and only 20 or 40 dB per decade roll-off above that. Ideally, a higher sample rate would be used. Also, a low-pass filter ideally removes signals above the corner frequency, not below the corner frequency as stated in the text.

The filters on the SR560 preamplifier have a 6dB per octave roll-off bandpass filter (<https://www.thinksrs.com/products/sr560.html>), and the sampling rate was limited to 2Msps based on disk speed dump limitations on the ADC. To change the measurement sampling rate, would require a different measurement system altogether which was not available. Since we used a careful bandpass filtering scheme to reduce the risk of preamplifier saturation, and protected from electric only mixing confounds with a two-tone electrical mixing test **Supplement 9**, **Figure 5** acoustic isolation test, and **Figure 2b** to separate acoustoelectric mixing from electric frequency mixing, we considered these three tests in conjunction with the front-end filtering scheme which limits saturation risks, sufficient to eliminate electric only mixing as the cause of the sum and difference frequency products.

138 – corrected to state above the corner frequency.

145. The measurement used for the noise signal should be specified (VP-P? VRMS?). The carrier artifact amplitudes in Figure 3d appear strangely clustered into three groups, but the DC offsets in Figure 3e do not. Some discussion here might be helpful.

The SNR method is described in Figure 3 h legend which also describes how the noise is calculated: ‘h, SSVEP signal-to-noise ratio (dB) comparison of continuous and no ultrasound 10Hz visual evoked potential paradigm. (r^2) = 0.60, $P = 0.54$; US group (mean \pm s.d. = **20.91 \pm 14.55**); No US group (mean \pm s.d. = **19.05 \pm 7.73**) n=30. SNR signal defined as ASD=10Hz compared to noise defined as mean of surrounding ± 5 spectral bins.’

The carrier artefact amplitudes tended to be the same amplitude throughout an experiment, leading to the appearance of clusters. This amplitude changed based on the height of the transducer above the mouse, and sometimes that height changed by a millimetre in which case there was a smaller difference in amplitudes.

The DC offsets in **Figure 3 e** are not directly proportional to the carrier amplitudes as they are not caused by rectification of the 500kHz carrier wave.

184. It would be helpful to know if the rat's eyes were momentarily covered as a quick

check in the acoustoelectric setup, similar to the verification done for the VEP measurements in Supplemental Note 4.

The LED blocking confound test was not repeated in every experiment, although the same hardware, cables and mouse configuration was used as the LED blocking confound test shown in **Supplemental Note 6** in every experiment. Unless there were hardware or mouse configuration changes between experiments, we should expect the result to be the same. Each mouse experiment was time bound as the evoked neural signal amplitudes decreased over time under Isoflurane anaesthesia, meaning not all confound tests could be performed in the same experiment.

To reduce the risk of electric only mixing from the LED, we performed electric only mixing confound tests in **Figure 2b**, **Figure 5** and **Supplement 9** which can separate acoustoelectric frequency mixing from electric only frequency mixing.

410. In Figure 6C, the vertical scale for the measured neural signal should be listed.

To compare the two signals both visually and as an input to the Pearson correlation, both signals were normalized as this ensures that the relative amplitudes of the original and recovered signals are not relevant – only the correlation. Hence, listing it as normalized enables both signals to be both visually compared for similarity which is then further backed up with the quantitative Pearson correlation which requires normalization of the input signals before computation.

Figure 4 d shows the SSVEP amplitude compared against the demodulated recovered signal amplitude.

426. Minor point – the $4\text{nV}\sqrt{\text{Hz}}$ figure only applies if the recording electrodes are very low impedance and only at the upper end of the amplifier passband. The noise floor in the lower frequency band will be higher due to $1/f$ noise in the amplifier and higher impedances in the electrodes at low frequencies.

We took the $4\text{nV}\sqrt{\text{Hz}}$ noise figure from the SR560 preamplifier specification which was a measurement made at 1kHz - <https://www.thinksrs.com/products/sr560.html> and have now added that the noise figure was measured at 1kHz into the methods.

Agreed that the thermal noise follows $1/f$ and will be present in both the mouse and any instrumentation.

504. The full details and parameters of the of the filtering and demodulation methods should be described or made available for groups wishing to replicate the work. Supplementary Figure S2 seems to be labeled as S8.

The filters used are the ones which ship with the SR560 from Stanford Research Systems. They are fully documented in the SR560 manual with full circuit schematic and a parts list in the manual located here - <https://www.thinksrs.com/products/sr560.html>

The demodulation is fully described in the **Methods** with example data file included with the article under the data and code availability sections, located on figshare: <https://figshare.com/s/2ca17d7ca823aaa59748>.

Supplementary Figure 2 has been relabelled.

Supplementary Figure S2e, the caption says the amplifier output is set to 40Vpp, and the pressures peak at 1.75 MPa, but in Fig S2, at 40Vpp the pressure is reported as ~1 MPa.

Thank you for pointing this out, the figure legend is updated appropriately.

Reviewer #5:

I co-reviewed this manuscript with one of the reviewers who provided the listed reports. This is part of the Communications Engineering initiative to facilitate training in peer review and to provide appropriate recognition for Early Career Researchers who co-review manuscripts.

Reviewer #6:

I co-reviewed this manuscript with one of the reviewers who provided the listed reports. This is part of the Communications Engineering initiative to facilitate training in peer review and to provide appropriate recognition for Early Career Researchers who co-review manuscripts.

References:

1. Blackmore, J., Shrivastava, S., Sallet, J., Butler, C. R. & Cleveland, R. O. Ultrasound Neuromodulation: A Review of Results, Mechanisms and Safety. *Ultrasound Med Biol* **45**, 1509–1536 (2019).
2. Rintoul, J. L., Neufeld, E., Butler, C., Cleveland, R. O. & Grossman, N. Remote focused encoding and decoding of electric fields through acoustoelectric heterodyning. *Communications Physics* **2023 6:1 6**, 1–11 (2023).
3. Leung, S. A. *et al.* Transcranial focused ultrasound phase correction using the hybrid angular spectrum method. *Sci Rep* **11**, 1–13 (2021).
4. Zhuang, Y. & Chen, D. Accurate spectral testing with non-coherent sampling for large distortion to noise ratios. *Proceedings of the IEEE VLSI Test Symposium 2016-May*, (2016).
5. Braun, V., Blackmore, J., Cleveland, R. O. & Butler, C. R. Transcranial ultrasound stimulation in humans is associated with an auditory confound that can be effectively masked. *Brain Stimul* **13**, 1527 (2020).
6. Froudarakis, E. *et al.* The Visual Cortex in Context. *Annu Rev Vis Sci* **5**, 317 (2019).
7. Martin, E. *et al.* IIRUSST consensus on standardised reporting for transcranial ultrasound stimulation. *Brain Stimul* **17**, 607–615 (2024).

8. Alvarez, A., Preston, C., Trujillo, T. & Witte, R. S. Acoustoelectric imaging for beat-to-beat cardiac activation wave mapping in an in vivo swine model. *IEEE International Ultrasonics Symposium, IUS 2020-September*, (2020).
9. Fda. Information for Manufacturers Seeking Marketing Clearance of Diagnostic Ultrasound Systems and Transducers. *Ultrasound* 1–64 (2008).
10. Martin, E. *et al.* Ultrasound system for precise neuromodulation of human deep brain circuits. *Nature Communications* 2025 16:1 **16**, 1–14 (2025).
11. Buzsáki, G., Anastassiou, C. A. & Koch, C. The origin of extracellular fields and currents-EEG, ECoG, LFP and spikes. *Nat Rev Neurosci* **13**, 407–420 (2012).
12. Jobst, B. C. & Cascino, G. D. Resective Epilepsy Surgery for Drug-Resistant Focal Epilepsy: A Review. *JAMA* **313**, 285–293 (2015).
13. Boyden, E. S., Zhang, F., Bamberg, E., Nagel, G. & Deisseroth, K. Millisecond-timescale, genetically targeted optical control of neural activity. *Nat Neurosci* **8**, 1263–1268 (2005).
14. Jobst, B. C. & Cascino, G. D. Resective Epilepsy Surgery for Drug-Resistant Focal Epilepsy: A Review. *JAMA* **313**, 285–293 (2015).
15. Alonso, P., Cuadras, D., Gabriëls, L. & Denys, D. Deep Brain Stimulation for Obsessive-Compulsive Disorder: A Meta-Analysis of Treatment Outcome and Predictors of Response. *PLoS One* 1–16 (2015) doi:10.1371/journal.pone.0133591.
16. Holtzheimer, P. E. & Nemeroff, C. B. Advances in the treatment of depression. *NeuroRx* **3**, 42–56 (2006).
17. Gilden, D. L., Thornton, T. & Mallon, M. W. 1/f Noise in Human Cognition. *Science (1979)* **67**, 1837–1839 (1995).
18. Gerster, M. *et al.* Separating Neural Oscillations from Aperiodic 1/f Activity: Challenges and Recommendations. *Neuroinformatics* **20**, 991–1012 (2022).
19. Mandija, S., Petrov, P. I., Vink, J. J. T., Neggers, S. F. W. & van den Berg, C. A. T. Brain Tissue Conductivity Measurements with MR-Electrical Properties Tomography: An In Vivo Study. *Brain Topogr* **34**, 56 (2020).
20. Hao, J. J. *et al.* Simulation of microwave propagation properties in human abdominal tissues on wireless capsule endoscopy by FDID. *Biomed Signal Process Control* **49**, 388–395 (2019).
21. Ouyang, G., Hildebrandt, A., Schmitz, F. & Herrmann, C. S. Decomposing alpha and 1/f brain activities reveals their differential associations with cognitive processing speed. *Neuroimage* **205**, 116304 (2020).
22. Vergani, A. A. Hans Berger (1873–1941): the German psychiatrist who recorded the first electrical brain signal in humans 100 years ago. *Adv Physiol Educ* **48**, 878–881 (2024).
23. Sadleir, R. J. & Argibay, A. Modeling skull electrical properties. *Ann Biomed Eng* **35**, 1699 (2007).
24. Cohen, D. Magnetoencephalography: Evidence of Magnetic Fields Produced by Alpha-Rhythm Currents. *Science (1979)* **161**, 784–786 (1968).
25. Glover, G. H. Overview of Functional Magnetic Resonance Imaging. *Neurosurg Clin N Am* **22**, 133 (2011).

26. Lavandier, B., Jossinet, J. & Cathignol, D. Quantitative assessment of ultrasound-induced resistance change in saline solution. *Med Biol Eng Comput* **38**, 150–155 (2000).
27. Jossinet, J., Lavandier, B. & Cathignol, D. The phenomenology of acousto-electric interaction signals in aqueous solutions of electrolytes. *Ultrasonics* **36**, 607–613 (1998).

Reviewers' comments:

Dear reviewers,

We want to thank you for your insightful assessments and support of our manuscript.

Reviewer 4: "The novelty of this paper is that it is the first time this will be done in vivo in a rodent model with EEG-level signals. The authors present methods and tests to reduce artifacts and to verify that the signals recorded were created by acoustoelectric effect. The authors have made good efforts to try to validate their methods and demonstrate that they were generating and recovering biopotential-related acoustoelectric signals. They were also forward in disclosing their difficulties managing artifacts and noise problems in doing their measurements and explaining their methods for working through them."

In the sections below, we respond (in blue), point by point, to your comments (in black).

Sincerely,

Jean Rintoul

Reviewer #4:

While the authors have added analysis to show that the DC artifact "is not due to neural response", the bigger concern is not that the DC artifact is neural in nature, but that the DC offset suggests that the 500kHz artifact is somehow being partially rectified or demodulated through a non-linearity somewhere in the system (perhaps in the amplifier's electronics), creating the potential for what the authors call "electrical only mixing". These and other possible non-acoustoelectric mixing mechanisms are still difficult to convincingly rule out in the current work. However, the two-tone, off-axis, and other tests performed do show reasonable effort to mitigate these concerns and are clearly described.

A sentence was added to the discussion:

To simplify signal recovery, we employed continuous-wave ultrasound, which facilitated frequency demodulation but also produced a DC-offset artefact that could be partially suppressed with high-pass filtering. Further systematic investigations in phantoms will be essential to determine the underlying cause of this artefact and develop methods for its mitigation.'

The revisions to Figure 1 are more accurate, but they still do not match the visual and motor cortex placements described in the text, show the ultrasound focus point, or use anatomically correct diagrams. The figure is labeled as "concept", but it still creates the impression that this configuration with electrodes away from the focus is used in the current work.

Figure 1 has been updated so that the ultrasound focus is closer to the visual cortex, with the electrode located nearby. An addition has been made to the caption to specify that 'The depicted positions of the ultrasound focus and electrodes are illustrative and do not reflect the precise experimental configuration in the visual cortex.'

The Discussion has been successfully updated to cover some of the potential safety challenges for translation to humans, but it does not address the much bigger challenges of signal amplitudes involved and how this is expected to be feasible.

The following paragraph has been updated in the discussion highlighting the smaller acoustoelectric signal amplitude, with further discussion on how the thermal noise floor decrease enables improved signal-to-noise ratios at high frequencies:

‘Although the modulated acoustoelectric signal is smaller in amplitude than EEG, a key advantage of this approach is that it selectively shifts activity from neurons within the ultrasound focus to higher frequencies, isolating it from surrounding sources and thereby enhancing spatial specificity. Furthermore, acoustoelectric neural recording has a different noise profile to EEG or MEG, with potential advantage that the thermal noise floor^{34,49} which decreases with $\frac{1}{r}$, is much lower around the modulated frequency compared to the low frequency neural signals (see **Supplementary Note 10**). Compared to EEG, the detection of the small acoustoelectric neural signals need not be limited to electrode position and size as the spatial locus is controlled by ultrasound, enabling larger electrodes to be used to average away any remaining noise^{34,50}.‘

The criticisms above regarding Figure 1 and the Discussion are part of a larger remaining concern - There is a fundamental gap between what the authors show, which is local sensing near the ultrasound focus, and the stated goals and claims of being able to record AE modulated signals with electrodes that are far from the focus or on the scalp. AE modulated ECoG signals will be very local, very small dipole signals in the relatively large volume conductor of the head. These signals will fall off rapidly with distance ($\sim 1/r^2$) and they will be further attenuated drastically by the skull. Without a convincing phantom test, simulation, or analysis to show that such signals would be reasonably recordable away from the focus or at the scalp, it is impossible to extrapolate or suggest from the current work that remotely or non-invasively recording brain signals within the skull is possible with AE techniques. This must be addressed if the authors want to keep claims that the current work provides a pathway to systems that could potentially “surpass EEG” or support non-invasive brain recording applications. Alternatively, the manuscript should be revised to limit claims of AE possibilities that are implied from the demonstrated results.

The sentence containing the phrase ‘surpass EEG’ has been replaced with: ‘Thus, acoustoelectric neural recording may address certain limitations of EEG, although it is likely to introduce challenges of its own.‘

A new analysis has been added in **Supplementary Note 10**, using the *in vivo* data in **Fig. 4** to demonstrate improvements in the signal-to-noise ratio of the acoustoelectrically modulated signal compared to the original neural signal due to the decrease in thermal noise.

‘A key advantage of acoustoelectric neural recording is that shifting neural signals to high frequencies reduces thermal noise and improves signal-to-noise ratio (SNR). Using the 8 Hz *in vivo* dataset (**Fig. 4**), we compared the noise floor in the neural band (5–25 Hz) with the modulated band (500 kHz \pm 5–25 Hz) across twenty 30s trials. Noise was significantly higher at low frequencies (0.632 ± 0.080) than in the modulated range (0.015 ± 0.003 ; **Fig. S10 a**; $t_{(15)} = -4.81$, $P = 2.71e-5$). Consistently, SNR was lower in the neural band (9.65 ± 12.80 dB) than at the modulated frequencies (27.36 ± 8.97 dB), corresponding to an approximately 50-fold improvement (**Fig. S10 b**). Thus, although acoustoelectric conversion efficiency is small, high-

frequency modulation confers a substantial SNR benefit. Whether this advantage persists through the human skull remains to be tested.'

Reviewers' comments:

Dear reviewers,

We want to thank you for your insightful assessments and support of our manuscript. We have updated the manuscript to ensure full compliance with the journal's formatting and policy requirements and have updated mouse images with an anatomically accurate photo of a mouse.

Sincerely,

Jean Rintoul, PhD

Summary

This study sets out to demonstrate the use of focused ultrasound to heterodyne and record neural activity signals at the brain surface. This phenomenon, known as the acoustoelectric effect, has been used in previous studies for imaging electrical dynamics, most notably by Witte and colleagues at the University of Arizona. The novelty of this paper is that it is the first time this will be done *in vivo* in a rodent model with EEG-level signals.

The authors present methods and tests to reduce artifacts and to verify that the signals recorded were created by acoustoelectric effects. They also present a novel setup in which they simultaneously record both evoked potentials and the acoustoelectric heterodyned signals through the same brain surface electrodes using a single wideband amplifier and data acquisition system. Finally, they compare the direct brain surface recordings to the signals recovered by demodulating the high-frequency acoustoelectric signals.

The direct evoked potential recordings and demodulated acoustoelectric signals presented appear compellingly similar, and this supports the case that they successfully acquired the same underlying signal through these two different modalities. However, artifacts, variability, noise, and other limitations in their data make it difficult to be sure that the data show acoustoelectric capture of microvolt signals and not some form of crosstalk in their setup. More validation is needed to establish this. The paper also makes claims about potential capabilities for focal recording of EEG signals with remote or surface electrodes and focused ultrasound, but these claims are highly speculative and not supported by the current work.

Major comments (general and by line number)

- Although the paper is well referenced, the phrasing and argumentation in many places are not consistent with traditional understandings of neuroscience and electrophysiology. For example, Hodgkin-Huxley is a model for describing membrane-level dynamics, not brain-level dynamics as stated by the authors. Also, while it is true that EEG is spatially limited by attenuation from the skull, the bigger challenge for higher resolution neural recording is that the signals become more localized as smaller populations of neurons are targeted. Even without the skull, the volume conduction of the head and scalp makes spatially specific neural recordings difficult without using smaller electrodes placed close to the neural tissue of interest. Additional editing by a subject expert in electrophysiology and/or neurophysiology would likely improve reception of the paper by those audiences.

- The proposed use of acoustoelectric heterodyning to frequency shift a small focal area of neural signals is very interesting and an exciting possibility. However, as with EEG-level neural signals themselves, the very small frequency-shifted signals created at the

ultrasound focus are also likely to prove difficult to detect at farther points across the volume conductor of the head and through the skull. The authors imply that this is what they are doing in Figure 1 and claim in their discussion, but all the test setups and their physics experiments work by placing the acoustoelectric detection electrode directly in the ultrasound focus, where the acoustoelectric signal is going to be at its maximum. Moreover, the holes created in the skull where the electrodes are placed may also be concentrating the VEP electrical gradient around the electrode, further enhancing a potential acoustoelectric signal. To make the claims that microvolt, biopotential-related acoustoelectric signals are detectable remotely or through the skull, the authors should at least provide some sort of simulation model to predict that this is achievable within the noise limits of electrodes and electronic amplifiers. A convincing bench model with recording electrodes outside of the ultrasound focus would be even better. It's not possible to claim that remote sensing is feasible based on the tests described in Figure S8.

- The authors have made good efforts to try to validate their methods and demonstrate that they were generating and recovering biopotential-related acoustoelectric signals. They were also forward in disclosing their difficulties managing artifacts and noise problems in doing their measurements and explaining their methods for working through them. However, the combined nature of the signal path, the complexity of the signal processing, and some of the data (such as the acoustic signal inducing DC shifts in the electrode recordings) still make it difficult to rule out that some other type of signal mixing is present in their setup and that the observed modulation is not truly based on acoustoelectric effects. In addition, all the recordings seem to have been made with the recording electrodes in the ultrasound field, raising the possibility of ultrasound interactions with the electrodes themselves. More bench validation of the analog signal processing chain and verification of the digital signal processing with simulated data and noise signals are needed to establish that the claimed acoustoelectric modulation and detection are correct.

72. More information should be provided regarding the size of the Pt/Ir electrodes and the geometry of their placement (distance apart?). Without that information, it's impossible to know the applied electric field gradient or reproduce the setup. In addition, in Figures 2a and S8b, it appears that the electric field is applied orthogonal to the ultrasound propagation direction, but in Figure S8c, it appears that the electric field is provided in parallel with the ultrasound beam. Moreover, the electrical setup used to apply this signal is relevant and should be described. Ideally, the V_E would be applied via an isolation transformer to create a floating voltage source in the saline.

73. The configuration of the recording electrodes for the acoustoelectric signals is not really described other than to say that they are Pt/Ir electrodes. Their size (impedance?) and arrangement should be described more clearly. In the supplemental material, it appears that the acoustoelectric detection electrode and the V_E electrodes are moved together on an assembly, but this is not described.
77. The “non-focal electric field” or electrical artifact from the transducer does appear to be “non-focal”, but it also appears to vary significantly with position and to have structure in Figure 2(b)iii. This is very strange and implies that something else is possibly going on besides electrical coupling. For example, is it possible that the ultrasound is somehow generating an electromechanical artifact at the charge double layer at the electrode-saline interface? Or causing capacitive microphonic signals at the insulation of the wires? If there were other carrier transduction mechanisms present in the test setup, it could help explain some of the variability seen in 2(b)iii.
81. Here again, the details of the excitation and recording electrodes are important to understanding the work and should be further described. The acoustoelectric signal is proportional to the gradient of the electric field (and the ionic current) where the ultrasound is focused. Without knowing the electrode geometry and spacing, it is impossible to know the geometry of the induced gradients and if the applied fields are good models for the electric fields and currents present in biological tissue.
109. It’s not clear if the authors did a Pearson correlation that was shifted in phase to find the maximum correlation point. A simple direct Pearson correlation measure is not a good index of fidelity or similarity for periodic time-domain signals. For example, for a sine wave, if the recorded signal is identical but 90 degrees out of phase with the reference signal, the correlation would be zero. A more conventional metric like THD and/or some kind of characterization of the added noise in the reconstructed signal would be more appropriate here.
144. The DC artifact created by the ultrasound is very problematic as it implies that the 500kHz artifact is somehow being rectified or demodulated through a nonlinearity in the signal path. As a result, any amplitude modulation of the recorded 500kHz carrier (even acoustoelectric signals) would likely be demodulated to baseband signals that are added to the recording. A simple test of this would be to slightly amplitude modulate the source for the 500kHz carrier at 10Hz and see if a 10Hz signal appears in the low-pass filtered signal. This should be investigated further as this artifact mechanism could impact the fundamental claims of the work.

208. The use of the air gap is a clever approach for verification of mechanism, but the final SNR metric used here is unconvincing and difficult to judge without knowing the variability of the background noise itself. It could be that the air gap simply adds lots of noise in some cases. Moreover, the images in Figures 5(b)iii and 5(c)iii show a clear, compelling example of the difference in mixing between connected vs. isolated cases, but the SNR values in Figure 5(f) appear to vary wildly by orders of magnitude. It also doesn't seem reasonable that the proposed SNR method can give reliable SNRs as low as -20 dB. Is it possible to apply a better SNR metric here? Or explain why so much of the SNR data don't appear to demonstrate the effect as well as the examples in 5(b)iii and 5(c)iii.
230. This paragraph shows effort by the authors for trying to eliminate possible non-acoustoelectric sources of modulation. However, the claims here are strongly in conflict with Figures 3(b) and 3(c) given the strong DC artifact signal created in the electrode recordings when the 500kHz ultrasound carrier is present. This is generally indicative of some form of nonlinear rectification of the AC interference into a DC signal. If the AC signal has amplitude modulation, this translates into the demodulated signal getting mixed into the signal path. Nonlinear interactions between the amplifier and the electrode electrochemical interface can sometimes be a source of these effects. This should be analyzed further and/or explained.
291. There are certainly noise and electrode impedance benefits to working with signals in the 500kHz range as opposed to 10Hz in biological tissue. However, the experiment in Figure S8 and the claim that high-frequency signals have improved transmission characteristics in saline (and the brain) suggest misunderstandings of electrode impedances and the propagation of voltages and currents in volume conductors. The results shown in Figure S8 are more likely explained by the higher impedance of partially polarizable electrodes like Pt and Pt/Ir at lower frequencies due to their predominantly capacitive interface impedance. When the stimulation electrodes are subjected to a 1 mV AC voltage source, much more current is driven into the tissue at 500kHz due to the lower impedance of the electrodes at 500kHz. This in turn leads to higher voltages being measured by the measurement electrodes. The observations here are likely due entirely to the impedance of the stimulation electrodes rather than anything related to the propagation of electrical signals through tissue itself at these frequencies.
294. The possibility that the acoustoelectric signal might be detectable by electrodes in other positions than the ultrasound focus area is interesting and worth investigating, but it is not supported by the current work and the erroneous conclusions drawn from

Figure S8. Claims of advantages over state-of-the-art methods like EEG should be tempered.

Minor comments (general and detailed by line number)

- There are editing-level mistakes and word omissions throughout the manuscript. The inclusion of high numbers of significant digits in the figure captions should be reviewed. The descriptions of the methods are distributed in the Results section, the Methods section, the figure captions, and the supplemental material in ways that are difficult and tedious to follow. These could be revised for better flow and clarity.
- 9. This is simply not correct. There are a wide variety of tools available for recording neural activity in deep brain areas with high spatial specificity used in both research and clinical practice (depth electrodes, sEEG electrodes, etc.). It seems the authors are trying to say instead that they would like to find **non-invasive** ways of doing focal recordings in deep brain areas.
- 43. Why was citation 16 experimentation limited by the understanding of the underlying physics? Mention of literature by Witte is referenced, but there is a more recent paper by the group from 2020 (Alvarez et al., Applied Optics, vol 59, iss 36, pp 11292-11300).
- 76. The parenthetical expressions for “sum” and “difference” appear swapped.
- 80. In Supplementary Note 2, Figure S8 g, h, and i, the amplitudes of the two sideband signals (Δf and Σf) significantly differ, even though it seems the acoustoelectric mechanism should create them identically. This implies that the amplifier passband might not be flat and/or some sort of error might be present in the signal processing. This should be discussed, especially if it could affect the demodulation process.
- 82. Although the authors describe their differential electrodes for recording, they never mention the grounding configuration in their saline bench or animal experiments. Although differential amplifiers measure the voltages between two inputs, the saline and animals in these experiments must somehow be held within the common-mode input range of the amplifier. Without a third grounding electrode for this, the subject is only weakly grounded to the amplifier through its input impedances, and recordings become very sensitive to common-mode (CM) noise sources and distortion as the recording subject floats around the CM input range of the amplifier or comes close to saturating the input amplifier stages. Approaching input saturation limits in the amplifier can also lead nonlinearities in the signal path.
- 135. The multi-unit and local field potential recordings shown in Figure S4(c) show that their setup is capable of recording good VEP responses.

138. For a 2 Msps sample rate, a first- or second-order 1MHz low-pass filter is insufficient to prevent Nyquist-related aliasing. The filter will only have 3 to 6 dB of rejection at the corner and only 20 or 40 dB per decade roll-off above that. Ideally, a higher sample rate would be used. Also, a low-pass filter ideally removes signals above the corner frequency, not below the corner frequency as stated in the text.
145. The measurement used for the noise signal should be specified (V_{P-P} ? V_{RMS} ?). The carrier artifact amplitudes in Figure 3d appear strangely clustered into three groups, but the DC offsets in Figure 3e do not. Some discussion here might be helpful.
184. It would be helpful to know if the rat's eyes were momentarily covered as a quick check in the acoustoelectric setup, similar to the verification done for the VEP measurements in Supplemental Note 4.
410. In Figure 6C, the vertical scale for the measured neural signal should be listed.
426. Minor point – the $4nV/\sqrt{Hz}$ figure only applies if the recording electrodes are very low impedance and only at the upper end of the amplifier passband. The noise floor in the lower frequency band will be higher due to $1/f$ noise in the amplifier and higher impedances in the electrodes at low frequencies.
504. The full details and parameters of the of the filtering and demodulation methods should be described or made available for groups wishing to replicate the work.

Supplementary Figure S2 seems to be labeled as S8.

Supplementary Figure S2e, the caption says the amplifier output is set to 40Vpp, and the pressures peak at 1.75 MPa, but in Fig S2, at 40Vpp the pressure is reported as ~1 MPa.

Comments for Authors

While the authors have added analysis to show that the DC artifact “is not due to neural response”, the bigger concern is not that the DC artifact is neural in nature, but that the DC offset suggests that the 500kHz artifact is somehow being partially rectified or demodulated through a non-linearity somewhere in the system (perhaps in the amplifier’s electronics), creating the potential for what the authors call “electrical only mixing”. These and other possible non-acoustoelectric mixing mechanisms are still difficult to convincingly rule out in the current work. However, the two-tone, off-axis, and other tests performed do show reasonable effort to mitigate these concerns and are clearly described.

The revisions to Figure 1 are more accurate, but they still do not match the visual and motor cortex placements described in the text, show the ultrasound focus point, or use anatomically correct diagrams. The figure is labeled as “concept”, but it still creates the impression that this configuration with electrodes away from the focus is used in the current work.

The Discussion has been successfully updated to cover some of the potential safety challenges for translation to humans, but it does not address the much bigger challenges of signal amplitudes involved and how this is expected to be feasible.

The criticisms above regarding Figure 1 and the Discussion are part of a larger remaining concern - There is a fundamental gap between what the authors show, which is local sensing near the ultrasound focus, and the stated goals and claims of being able to record AE modulated signals with electrodes that are far from the focus or on the scalp. AE modulated ECoG signals will be very local, very small dipole signals in the relatively large volume conductor of the head. These signals will fall off rapidly with distance ($\sim 1/r^2$) and they will be further attenuated drastically by the skull. Without a convincing phantom test, simulation, or analysis to show that such signals would be reasonably recordable away from the focus or at the scalp, it is impossible to extrapolate or suggest from the current work that remotely or non-invasively recording brain signals within the skull is possible with AE techniques. This must be addressed if the authors want to keep claims that the current work provides a pathway to systems that could potentially “surpass EEG” or support non-invasive brain recording applications. Alternatively, the manuscript should be revised to limit claims of AE possibilities that are implied from the demonstrated results.